# Small molecule modulation of protein corona for deep plasma proteome profiling

Ali Akbar Ashkarran[1,2,16], Hassan Gharibi [3,16], Seyed Amirhossein Sadeghi[4], Seyed Majed Modaresi [5], Qianyi Wang[4], Teng-Jui Lin [6], Ghafar Yerima [7], Ali Tamadon [7], Maryam Sayadi[8], Maryam Jafari[9], Zijin Lin[1], Danilo Ritz[10], David Kakhniashvili[11], Avirup Guha[12], Mohammad R. K. Mofrad [7], Liangliang Sun [4], Markita P. Landry [6,13,14], Amir Ata Saei [15] ✉ & Morteza Mahmoudi [1,2] ✉

The protein corona formed on nanoparticles (NPs) has potential as a valuable diagnostic tool for improving plasma proteome coverage. Here, we show that spiking small molecules, including metabolites, lipids, vitamins, and nutrients into plasma can induce diverse protein corona patterns on otherwise identical NPs, significantly enhancing the depth of plasma proteome profiling. The protein coronas on polystyrene NPs when exposed to plasma treated with an array of small molecules allows for the detection of 1793 proteins marking an 8.25-fold increase in the number of quantified proteins compared to plasma alone (218 proteins) and a 2.63-fold increase relative to the untreated protein corona (681 proteins). Furthermore, we discovered that adding 1000 μg/ml phosphatidylcholine could singularly enable the detection of 897 proteins. At this specific concentration, phosphatidylcholine selectively depletes the four most abundant plasma proteins, including albumin, thus reducing the dynamic range of plasma proteome and enabling the detection of proteins with lower abundance. Employing an optimized data-independent acquisition approach, the inclusion of phosphatidylcholine leads to the detection of 1436 proteins in a single plasma sample. Our molecular dynamics results reveal that phosphatidylcholine interacts with albumin via hydrophobic interactions, H-bonds, and water bridges. The addition of phosphatidylcholine also enables the detection of 337 additional proteoforms compared to untreated protein corona using a top-down proteomics approach. Given the critical role of plasma proteomics in biomarker discovery and disease monitoring, we anticipate the widespread adoption of this methodology for the identification and clinical translation of biomarkers.

The quest to comprehensively analyze the plasma proteome has become crucial for advancing disease diagnosis and monitoring, as well as biomarker discovery[1,2]. Yet, obstacles like identifying low-abundance proteins remain owing to the prevalence of high-abundance proteins in plasma where the seven most abundant proteins collectively represent 85% of the total protein mass[3,4]. Peptides from these high-abundance proteins, especially those of albumin, tend to dominate mass spectra impeding the detection of proteins with lower abundance.

To address this challenge, techniques such as affinity depletion, protein equalizer, and electrolyte fractionation have been developed to reduce the concentration of these abundant proteins, thereby facilitating the detection of proteins with lower-abundance[5–7]. Additionally, a range of techniques has been developed to enhance the throughput and depth of protein detection and identification, from advanced acquisition modes to methods that concentrate low-abundance proteins or peptides for liquid chromatography-mass spectrometry (LC-MS/MS) analysis[5,8–13]. For instance, in the affinity depletion strategy[14], affinity chromatography columns are used with specific ligands that bind to high-abundance proteins such as albumin, immunoglobulins, and haptoglobin. However, the cost and labor associated with such depletion strategies hamper their application for large cohorts. As another example, the salting-out technique[15] is used to add reagents (e.g., ammonium sulfate) to selectively precipitate high-abundance proteins, leaving the lower-abundance proteins in the supernatant. However, these methods can introduce biases in precipitating lower-abundance proteins as well, therefore, additional robust strategies are needed to ensure low-abundance proteins with high diagnostic potential are not missed in biomarker discovery studies. More details on the limitations of these strategies are presented elsewhere[16].

Recently, nanoparticles (NPs) have gained attention for their ability to support biomarker discovery through analysis of the spontaneously-forming protein/biomolecular corona (i.e., a layer of biomolecules, primarily proteins, that forms on NPs when exposed to plasma or other biological fluids)[5,17–25]. The protein corona can contain a unique ability to concentrate proteins with lower abundance, easily reducing the proteome complexity for LC-MS/MS analysis[5,17,22]. While the physicochemical properties of NPs do indeed influence the structure of their protein corona, it is generally observed that nanoscale materials exhibit different protein abundances compared to the original plasma protein composition[26]. In essence, most NPs have the potential to form a protein corona with distinct protein composition and abundance, differing from the native plasma proteins[26].

The application of single NPs for biomarker discovery has limitations in achieving deep proteome coverage, typically enabling the detection of only hundreds of proteins[27]. To enhance proteome coverage and quantify a higher number of plasma proteins, the use of a protein corona sensor array or multiple NPs with distinct physicochemical properties can be implemented. This approach leverages the unique protein corona that forms on each NP to increase proteome coverage, but carries the drawback of having to analyze multiple NP samples and needing to test many NP types to reach the desired depth[5,22,28]. In addition, the use of single NPs offers several advantages over multiple NPs, particularly in terms of commercialization and the regulatory complexities associated with multi-NP systems[29]. Additionally, utilizing a single type of NP can streamline the MS analysis process, reducing the time required to analyze large cohorts in plasma proteomics studies.

Small molecules native to human biofluids play a significant role in regulating human physiology, often through interactions with proteins. Therefore, we hypothesize that small molecules might influence the formation of the NP protein corona and serve to enrich specific proteins including biomarkers or low-abundance proteins. Recent findings have reported that high levels of cholesterol result in a protein corona with enriched apolipoproteins and reduced complement proteins, which is due to the changes in the binding affinity of the proteins to the NPs in the presence of cholesterol[30]. Accordingly, we hypothesized that small molecules endogenous to human plasma may affect the composition of the NP protein corona differently depending on whether these molecules act individually or collectively[31].

Our work presents an efficient methodology that harnesses the influence of various small molecules in creating diverse protein coronas on otherwise identical polystyrene NPs. Our primary hypothesis, corroborated by our findings, posits that introducing small molecules into plasma alters the manner in which the plasma proteins engage with NPs. This alteration, in turn, modulates the protein corona profile of the NPs. As a result, when NPs are incubated with plasma pre-treated with an array of small molecules at diverse concentrations, these small molecules significantly enhance the detection of a broad spectrum of low-abundance proteins through LC-MS/MS analyses. The selected small molecules include essential biological metabolites, lipids, vitamins, and nutrients consisting of glucose, triglyceride, diglycerol, phosphatidylcholine (PtdChos), phosphatidylethanolamine (PE), ʟ-α-phosphatidylinositol (PtdIns), inosine 5′-monophosphate (IMP), and B complex and their combinations. The selection of these molecules was based on their ability to interact with a broad spectrum of proteins, which significantly influences the composition of the protein corona surrounding NPs. For example, B complex components can interact with a wide range of proteins including albumin[32,33], hemoglobin[32], myoglobin[34], pantothenate permease[35], acyl carrier protein[36], lactoferrin[37], prion[38], β-amyloid precursor[39], and niacin-responsive repressor[40]. Additionally, to assess the potential collective effects of these molecules, we analyzed two representative "molecular sauces." Molecular sauce 1 contained a blend of glucose, triglyceride, diglycerol, and PtdChos, and molecular sauce 2 consisted of PE, PtdIns, IMP, and vitamin B complex.

Why did we choose polystyrene NPs for this study? Our team has extensive experience in analyzing the composition and profiles of the protein corona on various types of NPs, including gold[41–43], super-paramagnetic iron oxide[44–46], graphene oxide[47–49], iron-platinum[50], zeolite[51,52], silica[53,54], polystyrene[53,55–57], silver[58], and lipids[22,59,60]. In this study, we specifically selected highly uniform polystyrene NPs for two primary reasons: (i) polystyrene NPs have a protein corona that encompasses a broad spectrum of protein categories, including immunoglobulins, lipoproteins, tissue leakage proteins, acute phase proteins, complement proteins, and coagulation factors. This diversity is crucial for achieving wide proteome identification, which is essential for our research objectives and (ii) these particles are tested widely for numerous applications in nanobiomedicine: we[55–57] and other groups[61–65] have conducted extensive optimization, employing a wide range of characterizations, including MS, to analyze the protein corona of polystyrene NPs. This rigorous optimization ensures highly accurate and reproducible results.

Our findings confirm that the addition of these small molecules in plasma generates distinct protein corona profiles on otherwise identical NPs, significantly expanding the range of the plasma proteome that can be captured and detected by simple LC-MS/MS analysis. Notably, we discover that the addition of specific small molecules, such as PtdChos, leads to a substantial increase in proteome coverage, which is attributed to the unique ability of PdtChos to bind albumin and reduce its participation in protein corona formation. Therefore, PtdChos coupled with NP protein corona analysis can replace the expensive albumin depletion kits and accelerate the plasma analysis workflow by reducing processing steps. Furthermore, our single small molecule-single NP platform reduces the necessity for employing multiple NP workflows in plasma proteome profiling. This approach can seamlessly integrate with existing LC-MS/MS workflows to further enhance the depth of plasma proteome analysis for biomarker discovery.

## Results

### Protein corona and small molecules enable deep profiling of the plasma proteome

We assessed the effect of eight distinct small molecules, namely, glucose, triglyceride, diglycerol, PtdChos, PE, PtdIns, IMP, and vitamin B complex, on the protein corona formed around polystyrene NPs. The workflow of the study is outlined in Supplementary Fig. 1.

Commercially available plain polystyrene NPs, averaging 80 nm in size, were purchased. Each small molecule, at varying concentrations (10 μg/ml, 100 μg/ml, and 1000 μg/ml; we selected a broad range of small molecule concentrations to determine the optimal levels for maximizing proteome coverage), was first incubated with commercial pooled healthy human plasma at 37 °C for 1 h allowing the small molecules to interact with the biological matrix. The concentration of each small molecule was carefully adjusted to ensure that the final concentration in the combined molecular solutions was 10 μg/ml, 100 μg/ml, or 1000 μg/ml for each component, consistent with the concentration used for individual small molecules. Subsequently, NPs at a concentration of 0.2 mg/ml were introduced into the plasma containing small molecules or sauces and incubated for an additional hour at 37 °C with agitation. It is noteworthy that the NPs concentration was chosen in a way to avoid any protein contamination (which was detected at concentrations of 0.5 mg/ml and higher) in the protein corona composition, which may cause errors in the proteomics data[55,66]. These methodological parameters were refined from previous studies to guarantee the formation of a distinct protein corona around the NPs. Supplementary Fig. 2 offers further details on our methodologies, showcasing dynamic light scattering (DLS), zeta potential, and transmission electron microscopy (TEM) analyses for both the untreated NPs and those covered by a protein corona[67]. The untreated polystyrene NPs exhibited excellent monodispersity, with an average size of 78.8 nm a polydispersity index of 0.026, and a surface charge of −30.1 ± 0.6 mV. Upon the formation of the protein corona, the average size of NPs expanded to 113 nm, and the surface charge shifted to −10 ± 0.4 mV. TEM analysis further corroborated the size and morphology alterations of the NPs before and after protein corona formation (Supplementary Fig. 2).

To investigate how spiking different concentrations of small molecules can influence the molecular composition of the protein corona, samples were subjected to LC-MS/MS analysis for high-resolution proteomic analysis. While the analysis of plasma alone led to the quantification of 218 unique proteins, analysis of the protein corona formed on the polystyrene NPs significantly enhanced the depth of plasma proteome sampling to enable the quantification of 681 unique proteins. Furthermore, the inclusion of small molecules further deepened plasma proteome sampling to enable quantification of between 397 and up to 897 unique proteins, depending on the small molecules added to plasma prior to corona formation. When comparing the use of protein coronas, both with and without the inclusion of small molecules, to the analysis of plasma alone (Fig. 1a and Supplementary Data 1), there is a notable increase—approximately a threefold rise—in the number of proteins that can be quantified. The CVs of the number of quantified proteins between three technical replicates were generally less than 1.54% for all sample types (Supplementary Table 1).

Interestingly, the concentration of small molecules did not significantly affect the number of quantified proteins in a concentration-dependent manner; only a small stepwise reduction in the number of quantified proteins was noted with increasing concentrations of glucose and diglycerol. Cumulatively, the incorporation of small molecules and molecular sauces into the protein corona of NPs led to a significant increase in protein quantification, with a total of 1793 proteins identified, marking an 8.25-fold increase compared to plasma proteins alone. Specifically, the addition of small molecules resulted in the quantification of 1573 additional proteins compared to plasma alone, and 1037 more proteins than the untreated protein corona. Strikingly, spiking 1000 μg/ml of PtdChos increased the number of quantified proteins to 897 (1.3-fold of quantified proteins in untreated plasma), singlehandedly. This observation prompted a detailed investigation into the influence of PtdChos on plasma proteome coverage, which is elaborated in the following sections. It is noteworthy that the superior performance of PtdChos alone compared to

Molecular Sauce 1 could be attributed to interactions between the small molecules in the mixture, which may have lowered the effective concentration of PtdChos (for example, the interactions between PtdChos and triglycerides)[68,69]. Mass spectrometry workflow and the type of data analysis have a critical influence on proteomics outcomes in general[9,70–73], as well as in the specific field of protein corona research[56,57,74]. For instance, our recent study demonstrated that identical corona-coated polystyrene NPs analyzed by different mass spectrometry centers resulted in a wide range of quantified proteins, varying from 235 to 1430 (5.1 fold increase as compared to plasma alone)[56]. To mitigate the impact of these variables on the interpretation of how small molecules can enhance proteome coverage, we chose to report our data as fold changes in the number of quantified proteins relative to control plasma and untreated corona samples. This approach offers a more objective assessment of the role of small molecules in enhancing proteome analysis, minimizing the confounding effects of different workflows and data analysis techniques that may be employed by various researchers.

The distribution of normalized protein intensities for the samples is shown in Fig. 1b. The median value in the plasma group was notably higher than in the other samples, although the overall distribution did not differ significantly. In general, the proteomes obtained from protein corona profiles in the presence of small molecules showed a good correlation (generally a Pearson correlation above 0.6 for most small-molecule comparisons) demonstrating the faithful relative representation of proteins after treatment with different small molecules (Supplementary Fig. 3).

## Small molecules diversify the protein corona composition

We next investigated if the addition of small molecules would change the type and number of proteins detected by LC-MS/MS. Indeed, each small molecule and the molecular sauces generated a proteomic fingerprint that was distinct from untreated protein corona or those of other small molecules (Fig. 1c). Spiking small molecules led to the detection of a diverse set of proteins in the plasma. Interestingly, even different concentrations of the same small molecules or molecular sauces produced unique fingerprints. A similar analysis was performed for the 117 shared proteins across the samples (Fig. 1d). The Venn diagrams in Supplementary Fig. 4a, b show the number of unique proteins that were quantified in the respective group across all concentrations which were not quantified in the plasma or in the untreated protein corona. These results suggest that spiking small molecules into human biofluids can diversify the range of proteins that are identifiable in protein corona profiles, effectively increasing proteomic coverage to lower abundance proteins. Such an enrichment or depletion of a specific subset of proteins can be instrumental in biomarker discovery focused on a disease area. This feature can also be used for designing assays where the enrichment of a known biomarker is facilitated by using a given small molecule. As representative examples, a comparison of enriched and depleted proteins for molecular sauce 1 and 2 against the untreated protein corona is shown in Supplementary Fig. 4c, d, respectively (Supplementary Data 2). In certain cases, the enrichment or depletion was drastic, spanning several orders of magnitude. The enriched and depleted proteins for molecular sauces 1 and 2 were mapped to KEGG pathways and biological processes in StringDB (Supplementary Fig. 4c, d). While most of the enriched pathways were shared, some pathways were specifically enriched for a given molecular sauce. For example, systemic lupus erythematosus (SLE) was only enriched among the top pathways for molecular sauce 2. Therefore, the small molecules can be potentially used for facilitating the discovery of biomarkers for specific diseases, or for assaying the abundance of a known biomarker in disease detection.

Similar analyses were performed for all the small molecules and the volcano plots for the highest concentration of each molecule (i.e., 1000 μg/ml) are demonstrated in Supplementary Fig. 5

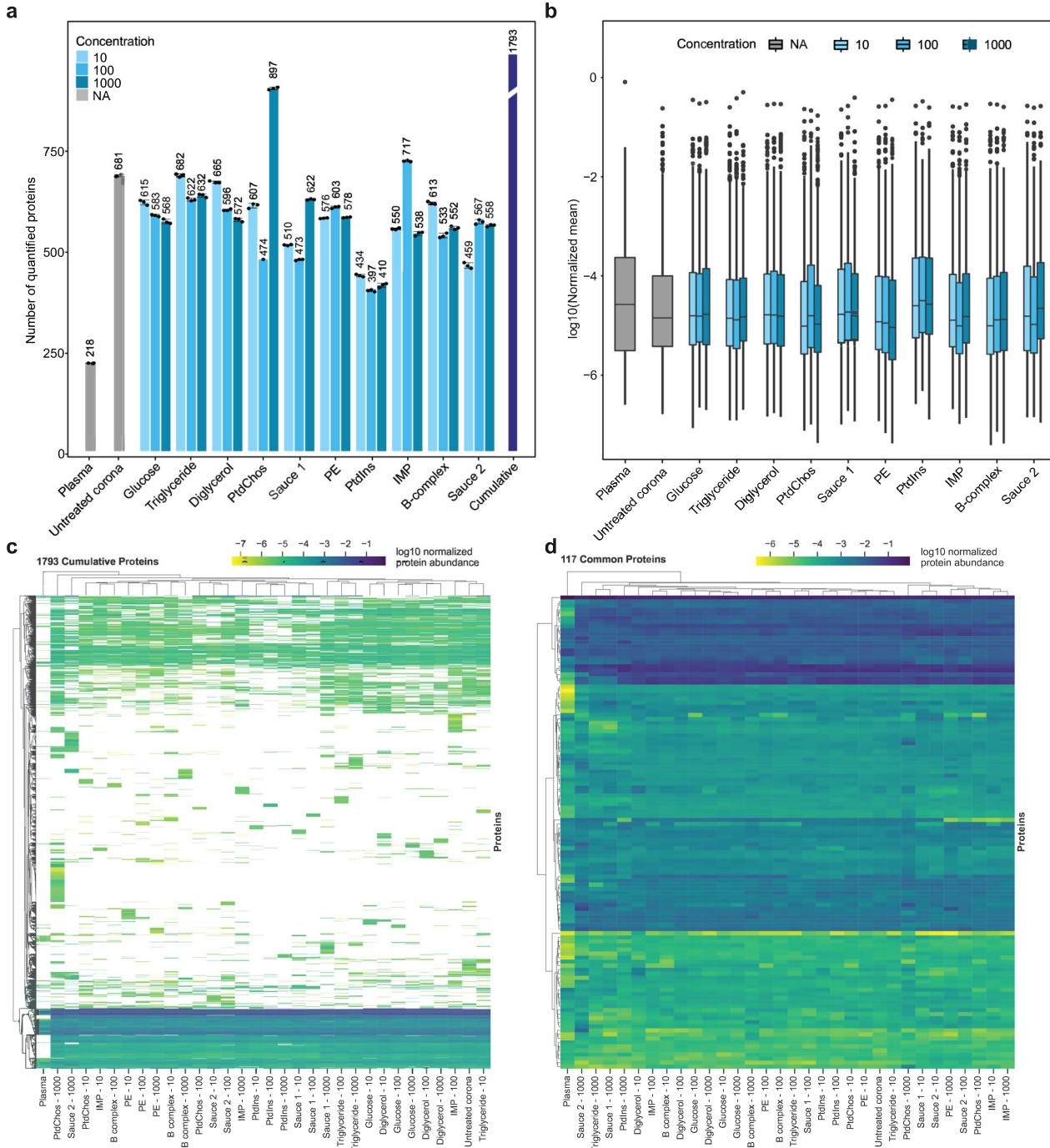

**Fig. 1 | Small molecules affect the plasma proteome sampling. a** The number of quantified proteins in plasma, untreated protein corona, and protein coronas in the presence of small molecules and molecular sauces (mean ± SD of three technical replicates). The cumulative number of unique proteins identified using untreated protein corona and corona treated with various small molecules is also shown using the purple bar. For a fair comparison, the database was performed individually for each small molecule (the higher the small molecule(s) concentration, the darker the blue shade). **b** The distribution of averaged normalized abundances of three technical replicates for proteins quantified in the plasma, untreated protein corona, and protein coronas in the presence of small molecules and molecular sauces (the higher the small molecule(s) concentration, the darker the blue shade; boxplot: center line, median; box limits contain 50%; upper and lower quartiles, 75% and 25%; maximum, greatest value excluding outliers; minimum, least value excluding outliers; outliers, more than 1.5 times of upper and lower quartiles). **c** Clustered heatmap of the normalized abundance of all 1793 proteins quantified across all samples. White denotes not detected. **d** Clustered heatmap of the normalized abundance of 117 shared proteins across all samples. Experiments were performed in three technical replicates. IMP inosine 5′-monophosphate, PE phosphatidy-lethanolamine, PtdIns L-α-phosphatidylinositol, PtdChos phosphatidylcholine.

(Supplementary Data 2). A pathway analysis was also performed for all the significantly changing proteins for each small molecule at all concentrations (Supplementary Fig. 6). To facilitate comparison, we have combined the enrichment analysis for all the samples vs the untreated protein corona in Supplementary Fig. 7

To demonstrate how small molecules affect the composition and functional categories of proteins in the protein corona, potentially aiding in early diagnosis of diseases (since proteins enriched in the corona are pivotal in conditions like cardiovascular and neurodegenerative diseases), we utilized bioanalytical methods[65] to categorize the

identified proteins based on their blood-related functions namely complement activation, immune response, coagulation, acute phase response, and lipid metabolism (Supplementary Fig. 8). In our analysis, apolipoproteins were major protein types that were found in the small molecule treated protein corona, and their types and abundance were heavily dependent to the type and concentrations of the employed small molecules (Supplementary Fig. 9). Similarly, the enrichment of other specific protein categories on NPs surfaces was influenced by the type and concentration of small molecules used (Supplementary Fig. 9). For example, antithrombin-III in coagulation factors plays a significant role in the protein corona composition of all tested small molecules, but this effect is observed only at their highest concentration. At lower concentrations, or in the untreated protein corona, this considerable participation is not evident (Supplementary Fig. 9). This ability of small molecules to modify the protein composition on NPs highlights their potential for early disease diagnosis (e.g., apolipoprotein in cardiovascular and neurodegenerative disorders)[31,75], where these protein categories are crucial in disease onset and progression[75].

## PtdChos reduces the plasma proteome dynamic range and increases proteome coverage by depleting the abundant plasma proteins

To understand whether the quantification of a higher number of proteins in protein corona profiles was due to a lower dynamic range of proteins available in human plasma for NP binding, we plotted the maximum protein abundance vs minimum protein abundance for plasma alone, and plasma-treated with small molecules in Supplementary Fig. 10. The plasma alone showed the highest dynamic range, suggesting that identification of low-abundance proteins would be most difficult from plasma alone. Conversely, the addition of small molecules was shown to reduce plasma protein dynamic range, thereby allowing for the detection of more peptides and quantification of proteins with lower abundance through the NP protein corona.

Notably, while albumin accounted for over 81% of our plasma sample, its representation was significantly lowered to an average of 29% in the protein coronas, both with and without small molecule modifications. This reduction was most pronounced with PtdChos treatment at 1000 μg/ml, where albumin levels dropped to around 17% of plasma proteins (Fig. 2a). Despite these changes, albumin remained the most abundant protein in all samples. A similar diminishing trend was observed for the second and third most abundant proteins, serotransferrin (TF) and haptoglobin (HB), which make up about 3.9% and 3.6% of plasma protein abundance, respectively. The rankings of these proteins' abundance in each sample are depicted above the panels in Fig. 2a. From this analysis, it is evident that the protein corona, both in its native form and when altered by small molecules, can drastically reduce the combined representation of the top three proteins from about 90% to roughly 29%. The most substantial reduction was observed with PtdChos at 1000 μg/ml, reducing the top three proteins' cumulative representation from 90% to under 17%. PtdChos treatment also effectively reduced the levels of the fourth most abundant plasma protein IGHA1. This significant decrease in the abundance of highly prevalent plasma proteins explains the marked increase in the number of unique proteins detected from NP corona samples treated with PtdChos (897 proteins identified in the PtdChos-treated protein corona vs 681 proteins identified in the untreated corona vs 218 proteins identified in the untreated plasma, as shown in Fig. 1a). These results indicate that high concentrations of PtdChos can be strategically employed to enable more comprehensive plasma protein sampling by specifically targeting and depleting the most abundant plasma proteins, especially albumin.

The stream (or alluvial) diagram in Fig. 2b shows the overall changes in the representation of proteins found in plasma upon incubation of protein corona with different concentrations of PtdChos. To validate this discovery, we prepared fresh samples treated with a series of PtdChos concentrations ranging from 100 μg/ml to 10,000 μg/ml (Supplementary Data 3). As shown in Fig. 2c, 957 proteins could be quantified in the protein corona treated with PtdChos at 1000 μg/ml, while neither lower concentration nor further addition of PtdChos did not enhance the number of quantified proteins. The CVs of the number of quantified proteins between three technical replicates were generally less than 2% for all sample types (Supplementary Table 2). The stream diagram in Fig. 2d shows the specific depletion of albumin and a number of other abundant proteins in plasma upon the addition of PtdChos, allowing for more robust detection of other proteins with lower abundance.

To confirm that the improved proteome coverage achieved with PtdChos treatment is independent of the LC-MS platform or the data acquisition mode used, we prepared new samples of plasma, untreated protein corona, and protein corona treated with 1000 μg/ml PtdChos, and analyzed them using LC-MS in the DIA mode. We identified 322 proteins in the plasma alone, 1011 proteins in the untreated protein corona samples, and 1436 proteins in the protein corona treated with PtdChos (1.4-fold increase over the untreated corona) (Supplementary Data 4). These findings not only validate the enhancement of plasma proteome coverage by PtdChos but also illustrate the capability of PtdChos to facilitate the in-depth profiling of the plasma proteome associated with protein corona formed on the surface of a single type of NP. Since the ratio of the number of quantified proteins through PtdChos spiking is generally around 1.4-fold higher than in the NP corona alone, PtdChos can be incorporated into any LC-MS workflow aiming to boost plasma proteome profiling. More optimized plasma proteomics pipelines, TMT multiplexing coupled to fractionation, or high-end mass spectrometers such as Orbitrap Astral are envisioned to quantify an even higher number of proteins than those reported in the current study.

To confirm the role of PtdChos in enhancing the proteome depth of the protein corona, we expanded our analysis by using additional NPs and four plasma samples from individual donors. Specifically, we tested seven additional commercially available and highly uniform NPs with distinct physicochemical properties: polystyrene NPs of varying sizes (mean diameters of 50 nm, 100 nm, and 200 nm) and surface charges (carboxylated and aminated polystyrene NPs, both with the mean diameter of 100 nm), as well as silica NPs with the mean sizes of 50 nm and 100 nm. These NPs have been extensively characterized and widely utilized for protein corona analysis by numerous research groups including our own[20,53,76-80].

The protein corona samples from different NPs were analyzed in the DIA mode with the 30 samples per day (SPD) setting with 44 min acquisition time. Our analysis revealed two key findings: (i) the physicochemical properties of NPs significantly influence the effectiveness of PtdChos in enhancing the number of quantified proteins in plasma, and (ii) incorporating additional plasma samples can markedly increase the overall number of identified proteins (Supplementary Data 5 and Supplementary Fig. 11a, b). Polystyrene NPs, in general, and due to their hydrophobic nature, showed higher protein detection capacity than silica NPs (p value = 0.012; Student's t-test, two-sided with unequal variance). The average number of quantified proteins using polystyrene NPs was 823.4 vs 633 with silica NPs, while cumulatively there were 1241 unique quantified proteins in polystyrene NPs compared to 1024 in silica. Polystyrene NPs with 200 nm size provided the highest proteome coverage, although the difference in the number of quantified proteins was comparable to the same type of NPs with other sizes. Plain and positively charged polystyrene NPs had a better performance than carboxylated NPs. Our analysis also revealed the inter-individual variabilities between patients. The percentage CVs of the number of proteins quantified across four donors were generally lower for polystyrene NPs than silica NPs (14.4 vs 21.6%) (Supplementary Table 3).

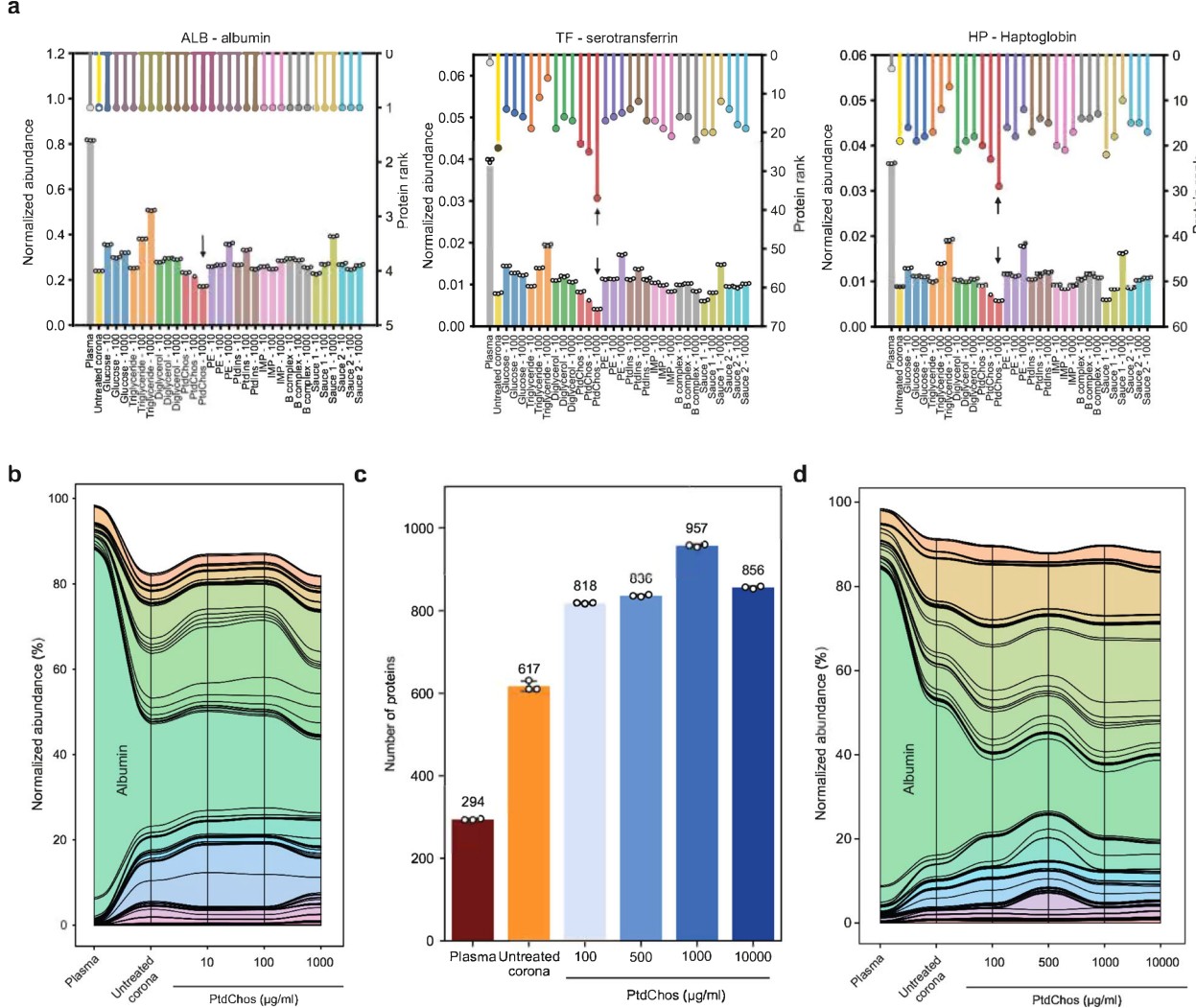

**Fig. 2 | PtdChos can deplete the most abundant plasma proteins in protein corona profiles. a** Normalized protein abundance (left axis, bar plot) and protein rankings (right axis, lollipop plot) in untreated plasma, untreated protein corona, and small-molecule treated protein corona. **b** A stream (or alluvial) diagram illustrating the significant depletion of abundant plasma proteins, particularly albumin, following the incubation of plasma with NPs and PtdChos (only shared proteins with plasma are included; colors are chosen randomly). **c** Total count of proteins identified in plasma, untreated protein corona, and protein corona treated with PtdChos at various concentrations (colors are chosen randomly; mean ± SD of three technical replicates). **d** A stream diagram demonstrating the depletion pattern of abundant plasma proteins, especially albumin, in response to NP addition and enhanced with escalating concentrations of PtdChos (colors are chosen randomly; only shared proteins with plasma are included). IMP inosine 5′-monophosphate, PE phosphatidylethanolamine, PtdIns ʟ-α-phosphatidylinositol, PtdChos phosphatidylcholine.

## PtdChos increases the number of detected plasma proteoforms

We then asked if PtdChos could enhance the number of detected proteoforms in top-down proteomics as well. Proteoforms represent distinct structural variants of a protein product from a single gene, including variations in amino acid sequences and post-translational modifications[81]. Proteoforms originating from the same gene can exhibit divergent biological functions and are crucial for modulating disease progression[82–84]. Therefore, proteoform-specific measurement of the protein corona, along with their improved detection depth through the use of small molecules, will undoubtedly provide a more accurate characterization of the protein molecules within the corona. We compared the chromatogram, the number of proteoform identifications, proteoform mass distribution, and differentially represented proteins between the untreated corona and PtdChos-treated samples. The LC-MS/MS data showed consistent base peak chromatograms, the number of proteoform identifications, and the number of proteoform-spectrum matches (PrSMs) across the technical triplicates of both the control and PtdChos-treated samples (Fig. 3a, b, respectively).

However, the treated sample exhibited a significant signal corresponding to the small molecule after 60 min of separation time (Fig. 3b), validating our hypothesis that small molecules interact with plasma proteins, causing the observed variation in the protein corona on the NPs' surface. Furthermore, the process of recovering intact proteins from the surfaces of NPs primarily collects proteins from the outer layer of the protein corona[85], as the inner layer is tightly bound to the NP surfaces through various physical and chemical forces[86]. This observation further confirms that PtdChos interacts with plasma proteins rather than directly with the NP surfaces, leading to the formation of its unique protein corona composition.

In total, 637 proteoforms were identified across the two samples (with technical triplicates for each sample) (Fig. 3c). Data analysis using Perseus software (Version 2.0.10.0) revealed that only 110 proteoforms overlapped between the two samples (the minimum number of valid values for filtering data was set to 1).

The proteoform mass distribution differed between the two samples (Fig. 3d). Although the average proteoform masses were

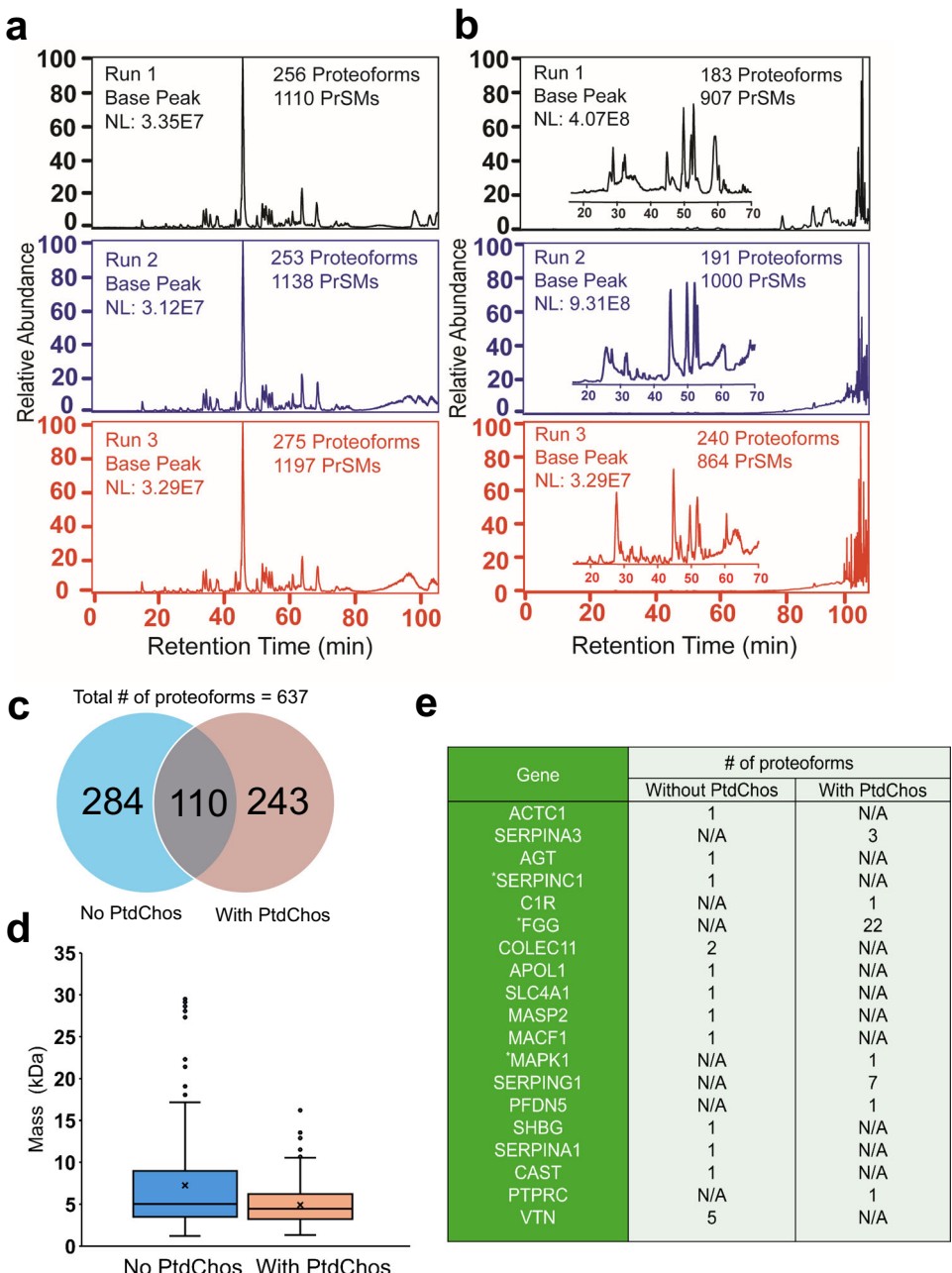

**Fig. 3 | Base peak chromatograms and proteoform analysis of protein corona samples. a** Base peak chromatogram of eluted protein corona without PtdChos and **b** with PtdChos in healthy human plasma after RPLC-MS/MS analyses. Two protein corona samples were prepared in parallel and analyzed by RPLC-MS/MS, with each sample measured in triplicate. **c** The number of proteoform identifications in each sample and the overlap of proteoform identifications between the two samples. **d** Mass distribution of proteoforms between the two samples, with the cross sign representing the mean proteoform mass in each sample (Centerline− median; box limits contain 50% of data; upper and lower quartiles, 75% and 25%; maximum−greatest value excluding outliers; minimum−least value excluding outliers; outliers−more than 1.5 times of the upper and lower quartiles). **e** Summary of some disease-related protein biomarkers identified by top-down proteomics. The Genes were determined according to the information in the Human Protein Atlas (https://www.proteinatlas.org/) and three genes labeled by "*" represent FDA-approved drug targets. NL normalization level, PrSMs proteoform-spectrum matches, PtdChos phosphatidylcholine.

similar, the box plot indicated a greater number of larger proteoform identifications in the control sample (over 20 kDa). We hypothesize that PtdChos can bind to large proteins, and due to the high concentration of PtdChos relative to the proteoforms, the signals of these large proteoforms may be obscured.

Additional data analyses identified differential proteins in this study (Fig. 3e). The top-down proteomics approach identified specific gene products that bind to the NP surface in the presence of PtdChos.

## PtdChos interacts with human serum albumin via hydrophobic interactions, H-bonds, and water bridges

To determine the types of interactions between albumin and PtdChos, we conducted all-atoms molecular dynamics (MD) simulations with various numbers of PtdChos molecules (Supplementary Fig. 12). First, we performed blind and site-specific molecular docking simulations to find the most favorable binding sites for PtdChos on albumin. We then used the top ten most favorable non-overlapping binding poses, as quantified by binding affinity, for our MD simulations (Supplementary

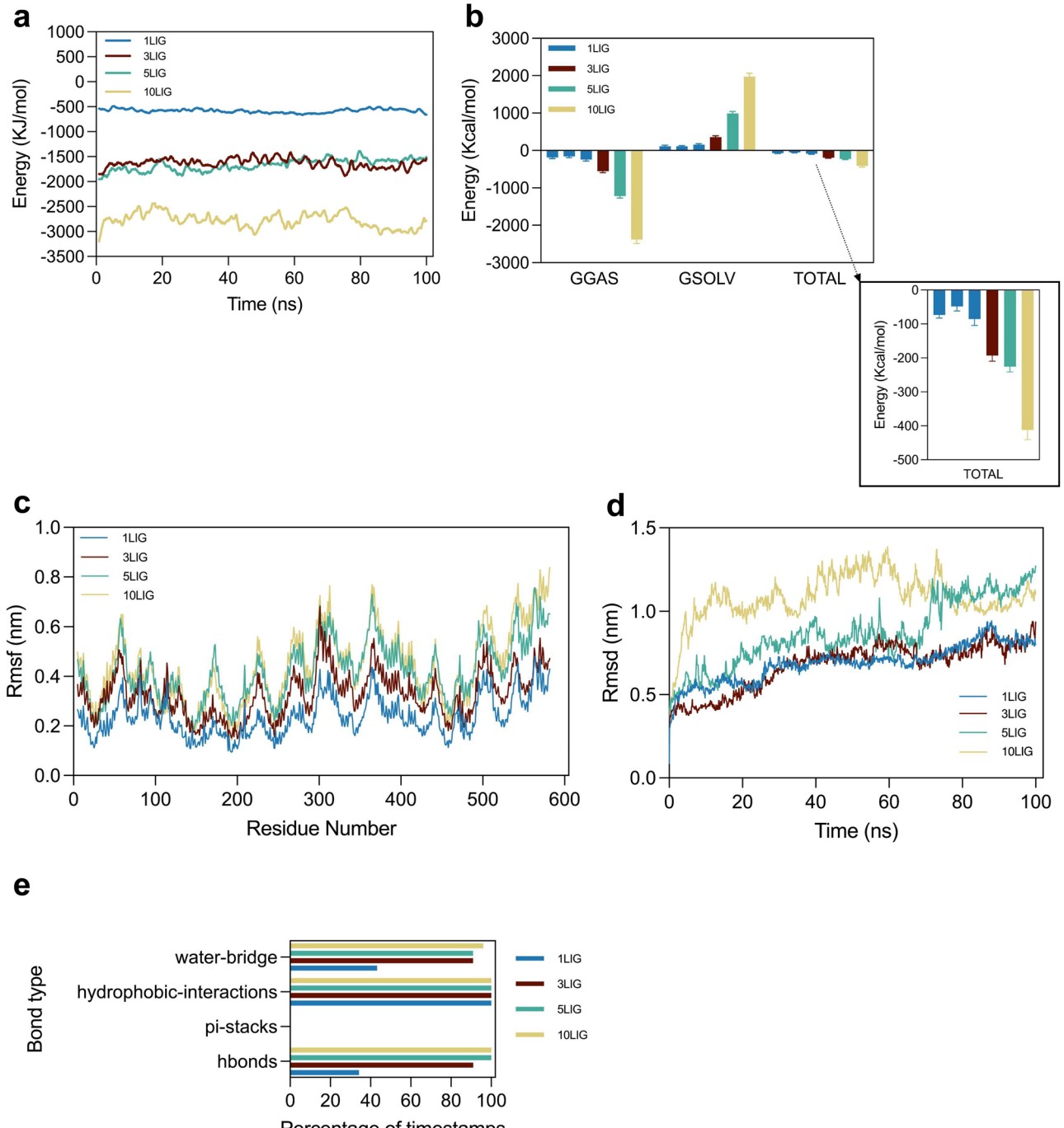

**Fig. 4 | PtdChos and albumin interaction analysis in all-atoms molecular dynamics simulations. a** Total linear interaction energies between albumin and various number of ligands systems over simulation time. The total energy represents the sum of Lennard-Jones and Coulombic energies. **b** Effective free energy of binding terms for the different systems over the entire simulation. From left to right, we have poses 1, 2, and 3 of the 1 ligand systems and the 3, 5, and 10 ligands systems. GGAS represents the energy of the gas phase, GSOLV, is the energy of solvation, and TOTAL is the sum of the two. For each simulation, the energies of 1000 frames were averaged, and the error bars show SD. **c** Average root mean square fluctuation of albumin residues for the four systems. **d** Root mean square deviation of PtdChos over time for the four types of systems. **e** Bond types present within each simulation. The *y*-axis represents the bond types, and the *x*-axis represents the percentage of simulation timestamps when each type of bond is present.

Fig. 12b, c). Four types of systems with the top 1, 3, 5, and 10 PtdChos molecules, respectively, were investigated via 100 ns simulations. As evidenced by the sum of Lennard-Jones and Coulombic interaction energies shown in Fig. 4a, PtdChos strongly interacts with albumin. A nearly additive effect occurs from 1 to 3 ligands added. However, the five ligands system has a similar total energy as the three ligands one. This may indicate that some PtdChos molecules do not strongly interact with albumin. When the number of ligands increased to 10, we noticed an almost 2-fold increase in energy as compared to the 5 ligands system. To further quantify the strength of interactions between albumin and PtdChos, we calculated the effective free energy of the four types of systems, obtaining a similar trend (Fig. 4b). The average root mean square fluctuations of albumin residues reveal consistent peaks with the increase in fluctuations as the number of ligands increases (Fig. 4c). This may suggest that the protein conformation does not change drastically based on the number of ligands

added. The average root mean square deviations of the PtdChos heavy atoms show similar values for the 1 and 3 ligands systems but higher values for 5 and 10 ligands systems (Fig. 4d). This confirms that the first few poses form a more stable interaction with albumin. Finally, Fig. 4e shows that albumin and PtdChos interact primarily via hydrophobic interactions, hydrogen bonding, and water bridges. The hydrophobic interactions formed between albumin and the long fatty acid chains are present throughout every simulation. On the other hand, the number of hydrogen bonds and water bridges increases significantly from the 1 ligand systems to the 3, 5, and 10 ligands systems. These interactions are mainly due to the phosphate group oxygen atoms.

## Discussion

The protein corona is a layer of proteins that spontaneously adsorbs on the surface of nanomaterials when exposed to biological fluids[55]. The composition and dynamic evolution of the protein corona is critically important as it can impact the interactions of NPs with biological systems (e.g., activate the immune system), can cause either positive or adverse biocompatibility outcomes, and can greatly affect NP biodistribution in vivo[55]. The specific proteins that adsorb on the surface of the NPs depend on various factors, including the physicochemical properties of the NPs and the composition of the surrounding biological fluid[56]. Metabolites, lipids, vitamins, nutrition, and other types of small biomolecules present in the biological fluid can interact with proteins in these fluids and influence their behavior, including their adsorption onto NPs. For example, it was shown that the addition of glucose and cholesterol to plasma can alter the composition of protein corona on the surface of otherwise identical NPs[30,67]. Small molecules can alter the protein corona of NPs, after interaction with plasma proteins, due to various mechanisms such as (i) their competition with proteins for binding to the surface of NPs; (ii) altering proteins' binding affinities to NPs; and (iii) changing protein conformation[30,67]. For example, previous studies revealed that triglyceride, PtdChos, PE, and PtdIns can interact with lipovitellin[87], C-reactive protein[88], protein Z[89], and myelin basic protein (MBP)[90], respectively. Each individual small molecule and its combinations interrogates tens to hundreds of additional proteins across a broad dynamic range in an unbiased and untargeted manner. Our results also suggest that endogenous small molecule function may help guide which small molecule(s) can enrich protein biomarkers of a specific disease class. Therefore, any changes in the level of the small molecules in the body can alter the overall composition of the protein corona, leading to variations in the types and number of proteins that bind to NPs and consequently their corresponding interactions with biosystems[30,31].

Among the various employed small molecules, we discovered that PtdChos alone demonstrates a remarkably high ability to reduce the participation of the most highly abundant proteins in protein corona composition. PtdChos is the most common class of phospholipids in the majority of eukaryotic cell membranes[91]. For a long time, it has been established that PtdChos can engage in specific interactions with serum albumin through hydrophobic processes[92,93], forming distinct protein–lipid complexes[92,94]. The results of our molecular dynamics evaluations of the interactions between PtdChos and albumin were in line with the literature. As a result, we found that the simple addition of PtdChos to plasma can significantly reduce albumin adsorption for the surface of polystyrene NPs, thereby creating unique opportunities for the involvement of a broader range of proteins with lower abundance in the protein corona layer. We also observed the same effects of PtdChos on enhancing the proteome coverage using different types of NPs. Not only is PtdChos an economical and simple alternative for conventional albumin depletion strategies, but it can also deplete several other highly abundant proteins as an added advantage. This approach reduces the necessity for employing NP arrays in plasma proteome profiling, and the cost and biases that can occur with albumin depletion. Additionally, PtdChos can help accelerate plasma analysis workflows by reducing the sample preparation steps.

Our top-down proteomics analysis of both untreated and PtdChos-treated protein coronas demonstrated that incorporating small molecules such as PtdChos can significantly enhance the quantification of proteoforms within protein corona profiles. Proteoforms, which are distinct structural variants of proteins arising from genetic variations, alternative splicing, and post-translational modifications, play a crucial role in determining protein functionality and are often closely linked to disease occurrence and progression[72,81,82,95,96]. By enriching the diversity and depth of proteoforms within the protein corona, the use of small molecules like PtdChos can substantially improve the level of information from plasma proteomics. This enhancement is particularly valuable for biomarker discovery, as the increased detection of proteoforms allows for a more nuanced understanding of disease mechanisms. The ability to capture a broader spectrum of proteoforms in the protein corona could lead to the identification of novel biomarkers that may otherwise be overlooked using traditional bottom-up proteomics approaches that mainly consider protein abundance[72].

Our study highlights the tremendous potential of leveraging small molecules to enhance the capabilities of protein corona profiles for broader plasma proteome analysis. By introducing individual small molecules and their combinations into plasma, we have successfully created distinct protein corona patterns on single identical NPs, thereby expanding the repertoire of attached proteins. Using our approach, we quantified an additional 1573 unique proteins that would otherwise remain undetected in plasma. This enhanced depth in protein coverage can be attributed, in part, to the unique interactions of each small molecule, allowing for the representation of a diverse set of proteins in the corona. Moreover, our findings underscore the influence of small molecules on the types and categories of proteins in the protein corona shell. This feature opens exciting possibilities for early disease diagnosis, particularly in conditions such as cardiovascular and neurodegenerative disorders, where enriched proteins, such as apolipoproteins, play pivotal roles. Importantly, our study demonstrated that PtdChos preferentially interact with highly abundant plasma proteins, thereby reducing their binding to NP surfaces. This reduction allows low-abundance proteins to contribute more significantly to the protein corona profile.

To further confirm the critical role of PtdChos in enhancing the depth of the plasma proteome, we employed the concept of actual causality, as outlined by Halpern and Pearl[97], rather than relying solely on correlation. This mathematical framework allowed us to substantiate how small molecules spiked into plasma can induce diverse protein corona patterns based on our proteomics results. Our findings revealed that among the small molecules tested, PtdChos was the actual cause of the observed increase in the proteomic depth of the plasma sample[98]. This effect was achieved by reducing the binding of highly abundant proteins and enhancing the representation of low-abundance proteins on the NP surfaces.

We acknowledge that the number of human plasma samples used in this study was limited, primarily due to our specific focus on improving proteome coverage through the use of a single pooled plasma sample. This approach effectively allows us to test and validate our hypothesis, given that the most abundant plasma proteins exhibit minimal variability between individuals. However, for future biomarker discovery applications, it is essential to expand the sample size to a more diverse cohort. This will ensure the platform fully accounts for biological variability and provides a more comprehensive and generalizable assessment of the proteome across different individuals.

One critical challenge that must be addressed is the standardization of proteomics analysis of the protein corona. Ensuring consistent and reproducible results across laboratories and core facilities is essential for the rapid development and successful translation of this

platform into clinical applications[56,57,74]. Addressing this challenge will require coordinated efforts from the scientific community to establish robust, universally accepted protocols. There are a few additional foreseeable limitations with the application of PtdChos. In certain scenarios, any depletion strategy could lead to distortion of the abundance of proteins in plasma, which can be mitigated by enforcing proper controls. Moreover, upon discovery of a biomarker, it can be validated in the cohort using orthogonal techniques such as Western blotting. Furthermore, similar to other albumin depletion strategies, certain proteins bound to albumin might be co-depleted (albuminome)[99].

In summary, our platform is capable of quantifying up to 1793 proteins when using a single NP with an array of small molecules, while only 218 and 681 proteins could be quantified using the plasma or the NP protein corona alone. We showed the possibility of quantifying up to 1436 proteins using a single NP and PtdChos alone using a single plasma sample. Similarly, in top-down proteomics, the addition of PtdChos to plasma prior to their interactions with NPs, can increase the number of quantified proteoforms in the protein corona. The cumulative number of detected proteins will therefore dramatically increase if this platform is applied to a cohort of patient samples with individual variability. Expectedly, with the progressive development of both top-down and bottom-up platforms[72], the depth of analysis can further increase toward the ultimate goal of achieving comprehensive human proteome coverage. Another alternative would be to combine our strategy with tandem mass tag (TMT) multiplexing and fractionation to achieve an even higher plasma proteome depth. We anticipate that this platform will find extensive applications in plasma proteome profiling, providing an unprecedented opportunity in disease diagnostics and monitoring.

## Methods
### Materials
Pooled healthy human plasma proteins, along with plasma from four individual healthy donors, were obtained from Innovative Research (www.innov-research.com) and diluted to a final concentration of 55% using phosphate buffer solution (PBS, 1×). Seven commercial NPs of various types (silica and polystyrene), sizes (50 nm, 100 nm, and 200 nm), and functional groups (plain, amino, and carboxylated) were sourced from Polysciences (www.polysciences.com). Small molecules were purchased from Sigma, Abcam, Fisher Scientific, VWR, and Beantown, and diluted to the desired concentration with 55% human plasma. Reagents for protein digestion, including guanidinium-HCl, DL-dithiothreitol (DTT), iodoacetamide (IAA), and trifluoroacetic acid (TFA), were obtained from Sigma Aldrich. Mass spectrometry-grade lysyl endopeptidase (Lys-C) was sourced from Fujifilm Wako Pure Chemical Corporation, and trypsin was obtained from Promega. Formic acid and C18 StageTips were purchased from Thermo Fisher Scientific.

### Protein corona formation on the surface of NPs in the presence of small molecules
For protein corona formation in the presence of small molecules, individual or pooled human plasma proteins 55% were first incubated with individual small molecules or in combination by preparing two molecular sauces of individual small molecules at different concentrations (i.e., 10 μg/ml, 100 μg/ml, and 1000 μg/ml) for 1 h at 37 °C. Then, each type of polystyrene NPs was added to the mixture of plasma and small molecules solution so that the final concentration of the NPs was 0.2 mg/ml and incubated for another 1 h at 37 °C. It is noteworthy that all experiments are designed in a way that the concentration of NPs, human plasma, and small molecules was 0.2 mg/ml, 55%, and 10 μg/ml, 100 μg/ml, and 1000 μg/ml, respectively. To remove unbound and plasma proteins only loosely attached to the surface of NPs, protein−NP complexes were then centrifuged at 14,000×*g* for

20 min, the collected NPs' pellets were washed three times with cold PBS under the same conditions, and the final pellet was collected for further analysis.

For the PtdChos concentration study, we used various concentrations of PtdChos (i.e., 250 μg/ml, 750 μg/ml, 1000 μg/ml, and 10000 μg/ml) and used the same protein corona method for the preparation of the samples for mass spectrometry analysis.

### NP characterization
DLS and zeta potential analyses were performed to measure the size distribution and surface charge of the NPs before and after protein corona formation using a Zetasizer nano series DLS instrument (Malvern company). A Helium-Neon laser with a wavelength of 632 nm was used for size distribution measurement at room temperature. TEM was carried out using a JEM-2200FS (JEOL Ltd) operated at 200 kV. The instrument was equipped with an in-column energy filter and an Oxford X-ray energy dispersive spectroscopy (EDS) system. Twenty microliters of the bare NPs were deposited onto a copper grid and used for imaging. For protein corona−coated NPs, 20 μl of samples was negatively stained using 20 μl uranyl acetate 1%, washed with DI water, deposited onto a copper grid, and used for imaging. PC composition was also determined using LC-MS/MS.

### Bottom-up LC-MS/MS sample preparation for the screening and concentration series experiments
The collected protein corona-coated NP pellets were resuspended in 20 μl of PBS containing 0.5 M guanidinium-HCl. The proteins were reduced with 2 mM DTT at 37 °C for 45 min and then alkylated with 8 mM IAA for 45 min at room temperature in the dark. Subsequently, 5 μl of LysC at 0.02 μg/μl in PBS was added and incubated for 4 h, followed by the addition of the same concentration and volume of trypsin for overnight digestion. The next day, the samples were centrifuged at 16,000×*g* for 20 min at room temperature to remove the NPs. The supernatant was acidified with TFA to a pH of 2−3 and cleaned using C18 StageTips. The samples were then heated at 95 °C for 10 min, vacuum-dried, and submitted to the core facility for LC-MS analysis.

LC-MS/MS Analysis: Dried samples were reconstituted with 1 μg of peptides in 25 μl of LC loading buffer (3% ACN, 0.1% TFA) and analyzed using LC-MS/MS. A 60-min gradient was applied in LFQ mode, with 5 μl aliquots injected in triplicate. Control samples (55% human plasma) were prepared with 8 μg of peptides in 200 μl of loading buffer and analyzed similarly. An Ultimate 3000RSLCnano (Thermo Fisher) HPLC system was used with predefined columns, solvents, and gradient settings. Data Dependent Analysis (DDA) was performed with specific MS and MS2 scan settings, followed by data analysis using Proteome Discoverer 2.4 (Thermo Fisher), applying the protocols detailed in our earlier publication (center #9)[56]. The PtdChos concentration series experiment was performed using the same protocol, and the samples were analyzed over a 120 min gradient.

### Sample preparation for top-down proteomics
Protein elution from the surface of NPs and purification were conducted based on procedures illustrated in our recent publications[85,100]. The protein corona-coated NPs (with/without PtdChos) were separately treated in a 0.4% (*w/v*) SDS solution at 60 °C for 1.5 h with continuous agitation to release the protein corona from the NP surface. Subsequently, the supernatant containing the protein corona in 0.4% SDS was separated from the NPs by centrifugation at 19,000×*g* for 20 min at 4 °C. To ensure thorough separation, the supernatant underwent an additional centrifugation step under the same conditions. The final protein corona sample was then subjected to buffer exchange using an Amicon Ultra Centrifugal Filter with a 10 kDa molecular weight cut-off, effectively removing sodium dodecyl sulfate (SDS) from the protein samples.

The buffer exchange process began by wetting the filter with 20 μl of 100 mM ABC (pH 8.0), followed by centrifugation at 14,000×*g* for 10 min. Next, 200 μg of proteins were added to the filter, and centrifugation was conducted for 20 min at 14,000×*g*. This step was repeated with the addition of 200 μl of 8 M urea in 100 mM ammonium bicarbonate, followed by centrifugation for 20 min at 14,000×*g*, and repeated twice to ensure complete removal of SDS and other small molecules. To eliminate urea from the purified protein, the filter underwent three additional rounds of buffer exchange. Specifically, 100 mM ABC was added to the filter, adjusting the final volume to 200 μl. All procedures were carried out at 4 °C to effectively eliminate urea from the protein corona.

Following buffer exchange, the total protein concentration was determined using a bicinchoninic acid (BCA) assay kit from Fisher Scientific (Hampton, NH), following the manufacturer's instructions. The samples were then stored overnight at 4 °C. The final protein solutions, consisting of 40 μl (without PtdChos initially) and 44 μl (with PtdChos initially) of 100 mM ABC with a protein concentration of 2.8 mg/ml, were prepared for LC-MS/MS analysis.

## Top-down proteomics LC-MS/MS

The RPLC separation was performed using an EASY-nLC™ 1200 system from Thermo Fisher Scientific. A 1-μL aliquot of the protein corona sample (0.3 mg/mL) was loaded onto a home-packed C4 capillary column (75 μm i.d. × 360 μm o.d., 20 cm in length, 3 μm particles, 300 Å, Bio-C4, Sepax) and separated at a flow rate of 400 nL/min. A gradient composed of mobile phase A (2% ACN in water containing 0.1% FA) and mobile phase B (80% ACN with 0.1% FA) was used for separation. The gradient profile consisted of a 105-min program: 0–85 min, 8–70% B; 85–90 min, 70–100% B; 90–105 min, 100% B. The LC system required an additional 30 min for column equilibration between the analyses, resulting in approximately 135 min per LC-MS analysis.

The experiments utilized a Q-Exactive HF mass spectrometer, employing a data-dependent acquisition (DDA) method. MS settings included 120,000 mass resolution (at *m/z* 200), 3 micro scans, a 3E6 AGC target value, a maximum injection time of 100 ms, and a scan range of 600–2000 *m/z*. For MS/MS analysis, parameters included 120,000 mass resolution (at 200 *m/z*), 3 micro scans, a 1E5 AGC target, 200 ms injection time, 4 *m/z* isolation window, and 20% normalized collision energy (NCE). During MS/MS, the top five most intense precursor ions from each MS spectrum were selected in the quadrupole and fragmented using higher-energy collision dissociation (HCD). Fragmentation occurred exclusively for ions with intensities exceeding 5E4 and charge states of 4 or higher. Dynamic exclusion was enabled with a 30-s duration, and the "Exclude isotopes" feature was activated.

## Top-down proteomics data analysis

Complex sample data were analyzed using Xcalibur software (Thermo Fisher Scientific) to obtain proteoform intensities and retention times. Chromatograms were exported from Xcalibur and formatted using Adobe Illustrator for the final figure presentation.

Proteoform identification and quantification were conducted using the TopPIC Suite (Top-down mass spectrometry-based Proteoform Identification and Characterization, version 1.7.4) pipeline[101]. Initially, RAW files were converted to mzML format using the MSConvert tool. Spectral deconvolution, which converted precursor and fragment isotope clusters to monoisotopic masses, and proteoform feature detection were performed using TopFD (Top-down mass spectrometry Feature Detection, version 1.7.4)[102]. The resulting mass spectra were stored in msalign files, while proteoform feature information was stored in text files.

Database searches were carried out using TopPIC Suite against a custom-built protein database (~2780 protein sequences), which included proteins identified in the BUP data. The search allowed for a maximum of one unexpected mass shift, with mass error tolerances of 10 ppm for precursors and fragments. Unknown mass shifts up to 500 Da were considered. False discovery rates (FDRs) for proteoform identifications were estimated using a target-decoy approach, filtering proteoform identifications at 1% and 5% FDR at the PrSM and proteoform levels, respectively.

Lists of identified proteoforms from all RPLC-MS/MS runs are provided in Supplementary Data 6. Label-free quantification of identified proteoforms was performed using TopDiff (Top-down mass spectrometry-based identification of Differentially expressed proteoforms, version 1.7.4) with default settings[103].

## LC-MS analysis by DIA

The samples were centrifuged at 14,000×*g* for 20 min to remove the unbound proteins. The collected NP pellets were washed three times with cold PBS under the same conditions. The samples were resuspended in 20 μl of PBS, and the proteins were reduced with 2 mM DTT (final concentration) for 45 min and then alkylated using 8 mM IAA (final concentration) for 45 min in the dark. Subsequently, 5 μl of LysC at 0.02 μg/μl was added for 4 h, followed by the same concentration and volume of trypsin overnight. The samples were then centrifuged at 16,000×*g* for 20 min at room temperature to remove the NPs then cleaned using C18 cartridges and vacuum dried.

Dried peptides were resuspended in 0.1% aqueous formic acid and subjected to LC-MS/MS analysis using an Exploris 480 mass spectrometer fitted with a Vanquish Neo (both Thermo Fisher Scientific) and a custom-made column heater set to 60 °C. Peptides were resolved using an RP-HPLC column (75 μm × 30 cm) packed in-house with C18 resin (ReproSil-Pur C18–AQ, 1.9 μm resin; Dr. Maisch GmbH) at a flow rate of 0.2 μl/min. The following gradient was used for peptide separation: from 4% B to 10% B over 7.5 min to 35% B over 67.5 min to 50% B over 15 min to 95% B over 1 min followed by 10 min at 95% B to 5% B over 1 min followed by 4 min at 5% B. Buffer A was 0.1% formic acid in water and buffer B was 80% acetonitrile, 0.1% formic acid in water.

The mass spectrometer was operated in DIA mode with a cycle time of 3 s. MS1 scans were acquired in the Orbitrap in centroid mode at a resolution of 120,000 FWHM (at 200 *m/z*), a scan range from 390 *m/z* to 910 *m/z*, normalized AGC target set to 300 %, and maximum ion injection time mode set to Auto. MS2 scans were acquired in the Orbitrap in centroid mode at a resolution of 15,000 FWHM (at 200 *m/z*), precursor mass range of 400 to 900, quadrupole isolation window of 7 *m/z* with 1 *m/z* window overlap, a defined first mass of 120 *m/z*, normalized AGC target set to 3000% and a maximum injection time of 22 ms. Peptides were fragmented by HCD with collision energy set to 28% and one microscan was acquired for each spectrum.

The acquired RAW files were searched individually using the Spectronaut (Biognosys v18.6) directDIA workflow against a *Homo sapiens* database (consisting of 20,360 protein sequences downloaded from Uniprot on 2022/02/22) and 392 commonly observed contaminants. Default settings were used.

For analysis of the impact of PtdChos treated plasma and different NPs, we chose a quicker LC-MS setup (30 SPD) consisting of an Exploris 480 fitted with an Evosep One using the following settings.

Dried peptides were resuspended in 0.1% aqueous formic acid, loaded onto Evotip Pure tips (Evosep Biosystems), and subjected to LC-MS/MS analysis using an Exploris 480 Mass Spectrometer (Thermo Fisher Scientific) fitted with an Evosep One (EV 1000, Evosep Biosystems). Peptides were resolved using a performance column−30 SPD (150 μm × 15 cm, 1.5 um, EV1137, Evosep Biosystems) kept at 40 °C fitted with a stainless-steel emitter (30 um, EV1086, Evosep Biosystems) using the 30 SPD method. Buffer A was 0.1% formic acid in water and buffer B was acetonitrile, with 0.1% formic acid.

The mass spectrometer was operated in DIA mode. MS1 scans were acquired in centroid mode at a resolution of 120,000 FWHM (at 200 *m/z*), a scan range from 350 *m/z* to 1500 *m/z*, AGC target set to

standard, and maximum ion injection time mode set to Auto. MS2 scans were acquired in centroid mode at a resolution of 15,000 FWHM (at 200 $m/z$), precursor mass range of 400–900 $m/z$, quadrupole isolation window of 12 $m/z$ without window overlap, a defined first mass of 120 $m/z$, normalized AGC target set to 3000% and maximum injection time mode set to Auto. Peptides were fragmented by HCD with collision energy set to 28% and one microscan was acquired for each spectrum.

The acquired RAW files were searched using the Spectronaut (Biognosys v19.0) directDIA workflow against a *Homo sapiens* database (consisting of 20,360 protein sequences downloaded from Uniprot on 2022/02/22) and 392 commonly observed contaminants. Default settings were applied except method evaluation was set to TRUE.

### In silico experiments
The crystal structure of human serum albumin (PDB code: 1AO6) was obtained from the protein data bank and used for all simulation setups (Supplementary Fig. 12). The structure of the PtdChos ligand was obtained from the CHARMM 36 force field files (Name: PLPC)[104].

### Molecular docking
Two blind docking methods and one site-specific docking were performed with Autodock Vina[105,106] software. The first blind docking used the whole albumin for the binding search and the second method consisted of multiple search boxes covering the entire albumin surface. The site-specific docking was performed based on crystallographic analysis of the binding sites on albumin for palmitic acid[107]. The top ten unique non-overlapping binding poses were kept for the subsequent molecular dynamics simulations.

### MD simulations
All-atom MD simulations were performed with GROMACS[108] free software and the CHARMM36[109] force field. Four types of protein-ligand systems were investigated (Supplementary Fig. 12b, c). Three 1 ligand systems, one 3 ligands, 5 ligands, and 10 ligands systems each were used for the simulations. The protein-ligand systems along with the TIP3P water model and a neutralizing salt concentration of 0.15 M NaCl were energy minimized using 5000 steps with an energy tolerance of 1000 KJ/mol/nm. The systems were subsequently equilibrated in 1 ns NVT and 4 ns NPT steps with a 1 fs timestep. The constant temperature for all runs was 310 K and the Berendsen pressure coupling was used. Production steps were then run for 100 ns with a 2 fs timestep with the Parrinello–Rahman barostat.

### Post-processing analysis
**Interaction energy.** The short-range nonbonded Coulombic and Lennard-Jones interaction energies between albumin and the ligands were calculated using GROMACS[108].

**Free energy calculation.** The free energy calculation for the entire 100 ns simulations was calculated using the gmx_MMPBSA[110,111] package with the generalized Born method. The entropic term was not considered. The residues and ligand atoms within 6 Å were selected for the calculation. The gas phase and solvation terms, as well as their sum, were averaged for 1000 frames and plotted for each simulation.

**RMSF.** Root mean square fluctuation per residue was calculated using GROMACS[108] free software after fitting to the first frame of the simulation. For the 1 ligand systems, all 3 simulation results were averaged.

**RMSD.** The root mean square deviation of the ligand(s) with respect to the energy-minimized structure was calculated using GROMACS[108]. The results of the three 1 ligand systems were averaged.

**Bond types.** The types of bonds formed between albumin and PtdChos were determined using the MD-Ligand-Receptor tool[112].

**Visualization.** All visualizations were made using the visual molecular dynamics (VMD) software[113].

### Data analysis
First, data were normalized by total protein intensity in each technical replicate. Then all the abundances were transformed into log10 and NA values were imputed by a constant value of −10 (in the heatmap figure). Except for PtdChos sample at 100 μg/ml, all samples were analyzed with three technical replicates. In the case, of DIA analysis of different NPs, there were four individual samples per group with no technical replicates. Statistical *t*-test with unequal variance were used to compare the differences between groups. Data analysis was performed using R (R version 4.1.0) with the help of ggplot2, dplyr, tidyr, ComplexHeatmap, and PerfromanceAnalytics packages.

### Statistics and reproducibility
All measurements were performed as a triplicate analysis of a given aliquot. The initial DIA analysis was performed in one replicate. The experiments on different NPs with PtdChos and DIA were performed on plasma samples from four individual donors.

### Reporting summary
Further information on research design is available in the Nature Portfolio Reporting Summary linked to this article.

## Data availability
The authors declare that all data supporting the findings of this study are available within the paper and its supplementary information and data files. The mass spectrometry data for all the bottom-up experiments are deposited in the database MassIVE with the identifiers MSV000094257. The MS RAW files for the top-down proteomics analysis were submitted to the ProteomeXchange Consortium through PRIDE[114] and assigned the dataset identifier PXD053359. The source data for Figs. 1 and 2a, b; and Supp Figs. 3, 4a, b, and 8–10 is Supplementary Data 1. The source data for Fig. 2c, d is Supplementary Data 3, and for Supplementary Figs. 4c, d and 5–7 is Supplementary Data 2. For Supplementary Fig. 11, the source data can be found in Supplementary Data 5, and for Fig. 4 the associated source is Supplementary Data 6.

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

## Acknowledgements

M.M. gratefully acknowledges financial support from the U.S. National Institute of Diabetes and Digestive and Kidney Diseases (grant DK131417). A.A.S. was supported by an Ambizione Fellowship from the Swiss National Science Foundation (SNSF grant number: PZOOP3_216203) and a grant from Karolinska Institutet (2-188/2022). L.S. thanks the support from the National Cancer Institute (NCI) through the grant R01CA247863. We acknowledge the support of a Burroughs Wellcome Fund Career Award at the Scientific Interface (CASI) (M.P.L.), a Dreyfus Foundation award (M.P.L.), the Philomathia Foundation (M.P.L.), an NSF CAREER award 2046159 (M.P.L.), an NSF CBET award 1733575 (to M.P.L.), a CZI imaging award (M.P.L.), a Sloan Foundation Award (M.P.L.), a McKnight Foundation award (M.P.L.), a Simons Foundation Award (M.P.L.), a Moore Foundation Award (M.P.L.), a Brain Foundation Award (M.P.L.), and a polymaths award from Schmidt Sciences, LLC (M.P.L.). M.P.L. is a Chan Zuckerberg Biohub San Francisco investigator.

## Author contributions

The concept, experimental design, and hypothesis development were done by (A.A.S. and M.M.). A.A.A. performed experimental procedures on the formation and analysis of protein corona; and sent the samples for bottom-up-proteomics analysis and different core facilities. H.G. analyzed the bottom-up proteomics data. S.A.S. and L.S. conducted and supervised (respectively) the protein corona analysis using top-down-proteomics. S.M.M., Q.W., T.J.L., M.S., M.J., Z.L., A.G., and M.L. contributed to literature search, data analysis, and/or discussion of the results. G.Y., A.T., and M.R.K.M. conducted and discussed molecular dynamics analysis. D.R. and D.K. conducted the bottom-up proteomics of the samples. All studies were supervised by A.A.S. and M.M.

## Funding

## Competing interests

Morteza Mahmoudi discloses that (i) he is a co-founder and director of the Academic Parity Movement (www.paritymovement.org), a non-profit organization dedicated to addressing academic discrimination, violence, and incivility; (ii) he is a co-founder of Targets' Tip and Albu-derm; and (iii) he receives royalties/honoraria for his published books,

plenary lectures, and licensed patents. The remaining authors declare no competing interests.

## Additional information

[1]Precision Health Program, Michigan State University, East Lansing, MI, USA. [2]Depatment of Radiology, College of Human Medicine, Michigan State University, East Lansing, MI, USA. [3]Division of Chemistry I, Department of Medical Biochemistry and Biophysics, Karolinska Institutet, Stockholm, Sweden. [4]Department of Chemistry, Michigan State University, East Lansing, MI, USA. [5]Biozentrum, University of Basel, Basel, Switzerland. [6]Department of Chemical and Biomolecular Engineering, University of California, Berkeley, Berkeley, CA, USA. [7]Molecular Cell Biomechanics Laboratory, Departments of Bioengineering and Mechanical Engineering, University of California Berkeley, Berkeley, CA, USA. [8]Department of Biomedical Engineering, Michigan State University, East Lansing, MI, USA. [9]Division of ENT Diseases, Department of Clinical Science, Intervention and Technology, Karolinska Institutet, Stockholm, Sweden. [10]Proteomics Core Facility, Biozentrum, University of Basel, Basel, Switzerland. [11]Proteomics and Metabolomics Core Facility, University of Tennessee Health Science Center, Memphis, TN, USA. [12]Cardio-Oncology Program, Medical College of Georgia at Augusta University, Augusta, GA, USA. [13]Department of Neuroscience, University of California, Berkeley, Berkeley, CA, USA. [14]Chan Zuckerberg Biohub, San Francisco, CA, USA. [15]Department of Microbiology, Tumor and Cell Biology, Karolinska Institutet, Stockholm, Sweden. [16]These authors contributed equally: Ali Akbar Ashkarran, Hassan Gharibi. ✉e-mail: amir.saei@ki.se; mahmou22@msu.edu

