## [Transparent Peer Review file · Nature Communications]

Small molecule modulation of protein corona for deep plasma proteome profiling

Corresponding Author: Professor Morteza Mahmoudi

Version 0:

Reviewer comments:

Reviewer #1

(Remarks to the Author)

The work presented in the submission titled, "Deep Plasma Proteome Profiling by Modulating Single Nanoparticle Protein Corona with Small Molecules" has a clearly defined hypothesis and definitely present some very interesting results. However, the authors are requested to work on the following suggestions to further revise the manuscript.

1. Abstract: The authors may include the specifics of small biomolecules here if possible. Also please indicate the specific concentration of phosphatidylcholine.

2. Introduction: In the second paragraph, can the authors cite limitations of specific methods separately. What about resolution in terms of no. of quantifiable proteins in the alternate existing methods.

In the third para, what about any specific size criteria and composition, do they generalize the nano sized particles for single NP strategy here?

In the Last paragraph of the introduction, can the authors clarify more on the basis of selection of small molecules. Is it based on previous findings or preliminary results.

Any specific rationale for selecting polystyrene NPs in this work. Can this advantage of small molecule-based analysis be leveraged for other types of particles having similar size range or composition.

3. Results:

In the first paragraph, how the authors select the composition of molecular sauce 1 and 2?

In the second para, whether charge of the bare NPs (Zeta potential) had any role to play in protein corona formation. Whether the pre incubation time of small molecules with the plasma proteins was optimized and determined at 1 h. Similarly the NP concentration of 0.2 mg/ml, how this was set? Since the authors presented a method development here, the steps need more clarity. Through they mentioned, "These methodological parameters were refined from previous studies", please indicate specific details.

The last line of the section titled, "Small molecules diversify the protein corona composition" can be moved to discussion.

4. Discussion:

Can the authors clarify on influence of competitive binding of proteins and selected small molecules to the NPs surface, and resolution of overall composition of protein in the corona?

The authors stated, "the simple addition of PtdChos to plasma can significantly reduce albumin's affinity for the surface of polystyrene NPs...". Whether relative affinity (binding constant) has a role to play here.

The results and analytical potential notwithstanding can the authors comment on the following points

(i) role of the competitive binding of small molecules and proteins from a mixture onto the NP surface, (ii) effect of pH on the

analytical process, (iii) whether this can be generalized to the other NPs and finally whether the size of the NPs have any specific role to play in the process, (iv) There are several studies reporting conformational changes in the plasma proteins by NPs (PS NPs too). Whether this would impact the analytical potential to develop newer diagnostics approaches?

Conclusions: The authors are encouraged to add a few lines on analytical challenges and limitations of the method in the conclusions section.

Reviewer #2

(Remarks to the Author)

In this article, Ashkarran et al. aim to build on prior research by exploring the utilization of the nanoparticle biomolecule corona for deep plasma proteomics analysis. The authors investigate how the addition or spiking of small molecules in plasma samples influences corona formation and subsequently impacts the number of plasma proteins identified by mass spectrometry. While the hypothesis presented in this work holds some novelty, there are major concerns and substantial experimental and data analysis work required for the manuscript to be considered for publication. Furthermore, given the primary focus on developing a proteomics pipeline without direct implications for biomarker discovery, the manuscript might be more suitable for publication in a proteomics-focused journal. Therefore, I would not recommend this manuscript for publication in Nature Communications.

Major Concerns:

- The underlying mechanism driving the hypothesis of this work remains vague, as the authors have not sufficiently addressed or experimentally investigated it. It is unclear whether the observed increase in the number of identified proteins is a result of the interaction between small molecules and the NP surface or a result of the competitive interaction of proteins with small molecules and NP surfaces. It is essential to understand whether small molecules interact with plasma proteins, with the NP surface, or potentially with both. Does for example PtdChos interact with albumin and other proteins in blood and subsequently reduces the interaction of highly abundant proteins with the NP surface? If this interaction occurs, the authors should provide evidence demonstrating the removal of PtdChos:protein complexes upon purification of the corona-coated NPs. Clarifying these aspects is crucial for establishing a robust mechanistic understanding and ensuring the validity of the hypothesis proposed by the authors.

- Selection of small molecules and 'personalized corona'. How were the types of small molecules and their combinations ('sauce') selected? Additionally, a significant limitation of this work is the use of pooled healthy plasma as the plasma source (three technical replicates). This experimental design fails to address individual variations in the plasma proteome. For instance, what if glucose levels are inherently elevated in an individual's plasma? Therefore, biological replicates (plasma samples obtained from various individuals) are necessary to demonstrate the efficiency the repeatability of the workflow proposed.

- The technical repeatability of the developed proteomics workflow is a significant concern in this work. Authors conduct only three technical replicates from pooled plasma, which may not be sufficient to assess the variability adequately. Calculating the coefficient of variation is essential to demonstrate the repeatability of the pipeline before proceeding to biological replicates.

- This work could have been significantly strengthened by incorporating additional experimental analyses using patient samples for plasma analysis. Such inclusion would have illustrated the advantages of the suggested workflow over previously published methodologies.

- One of the main findings in this work is the utilization of PtdChos to mitigate the interaction of NPs with highly abundant proteins, particularly albumin. It is not clear if authors suggest using PtdChos alone or in combination with other small molecules. In Figure 1a PtdChos-treated samples have the highest number of identified proteins. Could authors explain why it works better alone rather than in combination with other small molecules ('Molecular sauce 1')? Why is it important to discuss the cumulative number of unique identified proteins? Do authors suggest to aliquot plasma samples and separately analyse them with different small molecules? That would increase the steps/time/plasma volume required.

- Page 4, Authors mention: 'We note that our analysis focuses on the fold changes of quantified proteins compared to plasma and the untreated corona, rather than the absolute numbers of quantified proteins. This is mainly because the numbers of quantified proteins are strongly dependent to the workflow of mass spectrometry'.

I find this statement confusing and potentially misleading. While I agree that the number of identified proteins depends on the instrument settings and data analysis workflow used, the authors throughout the manuscript primarily focus the discussion on the number of identified proteins in comparison to previous work. Moreover, they employ two different experimental settings, instruments/facilities, digestion protocols, and data analysis workflows (Proteome Discoverer and R), which, in my opinion, weaken rather than strengthen the manuscript. Additionally, considering the emphasis given on the authors' previous work regarding mass spectrometry workflow biases, it would be beneficial for the authors to provide more clarity about the data analysis performed. The current paragraph on data analysis provides minimal information, and I believe more detail is necessary. Furthermore, it is unclear why only one replicate was performed for the DIA analysis.

Minor Concerns/comments:

- While employing a single type of NP can offer advantages over utilizing multiple types of NP surface modifications, the

authors should discuss why this specific type of NP was chosen. For example, is there a particular reason for using negatively charged NPs?

- Why were plasma samples diluted before protein corona formation? Authors should include dilution experiments, including the analysis of protein corona samples formed in undiluted plasma, to provide a comprehensive understanding of the effects of dilution on corona formation.

- While authors discuss some previous studies that utilize the NP biomolecule corona formation for proteomics analysis of plasma samples, they have neglected to cite relevant previously published work on this topic.

- In the introduction/discussion of the manuscript, the authors mention that plasma depletion methodologies are costly and time-consuming. How does the suggested workflow compare to immunodepletion methodologies in terms of cost and time efficiency? Corona formation and purification is also a multistep protocol.

- Page 4, Authors mention: 'Interestingly, the concentration of small molecules did not significantly affect the number of quantified proteins; only a small stepwise reduction in the number of quantified proteins was noted with increasing concentrations of glucose and diglycerol'. Could you please clarify where this data is presented?

- In my opinion, KEGG analysis does not add any important conclusion to this manuscript as healthy pulled plasma was used; I would understand it if plasma samples obtained from different individuals were analysed.

Reviewer #3

(Remarks to the Author)

In this manuscript, Ashkarran et al. reported an intriguing strategy to enhance the plasma proteome coverage through LC-MS/MS analysis. By incorporating small molecules into plasma, they significantly increased the diversity of protein coronas formed on nanoparticles (NPs), subsequently elevating the sensitivity for detecting proteins with lower abundances. This innovative and efficient methodology offers profound implications for deepening plasma proteome analysis, particularly in the realm of clinical biomarker discovery for disease diagnosis and monitoring. Nevertheless, several concerns ought to be addressed prior to publication:

1. The novelty of the study lies in using a single NP type and a small molecule to analyze a single plasma sample, setting a new standard in proteomic depth. However, to further enhance the rigor and reliability of the study, it is suggested that the authors provide more detailed information on the biological significance of the small molecules selected and the specific mechanisms of their interactions with proteins.

2. In this study, the majority of the smaller molecules employed were biological metabolites, chosen primarily due to their capacity to interact with plasma proteins. Consequently, it is expected that these metabolites would naturally be present in plasma. However, a crucial consideration lies in the background concentrations of these metabolites within the plasma. Did the authors factor in these endogenous concentrations when determining the pre-designed spiking concentrations? Upon examination of the results, it appears that the concentrations of these metabolites did not significantly influence the number of proteins identified for most molecules. This observation deserves further exploration and discussion, as it may provide insights into the complexity of interactions between metabolites and plasma proteins.

3. The concentrations of each molecules in the two "molecular sauces" should be specified.

4. The study found that the addition of 1000 µg/ml phosphatidylcholine (PtdChos) significantly increased the number of unique proteins within the protein corona (897 proteins). This finding has significant clinical translation potential. However, it is suggested that the authors further explore the impact of PtdChos concentration on other plasma proteins and whether this selective depletion mechanism affects the detection sensitivity of low-abundance proteins. More additional bioinformatics analysis on PtdChos-treated plasma samples were recommended to determine whether these changes affect the identification of potential biomarkers.

5. In the DDA analysis, the number of identified proteins in untreated corona and ptdchos (100 ug/uL) group were 681 and 474 (Figure 1a). In the DIA analysis, their counterparts were 617 and 836 (Figure 2c). Why DIA analysis could greatly improve the depth of identified proteins for ptdchos group but not for untreated corona group?

6. More details should be provided on the optimization of the data-independent acquisition (DIA) method, including the selection of acquisition parameters, data processing workflows, and how to ensure the accuracy and reproducibility of the data. Furthermore, authors should discuss the potential application of this method in different types of plasma samples and whether the detection of specific proteins can be optimized by adjusting the concentration of small molecules.

7. Proteomics analysis showed that Ptdchos efficiently reduce the abundance of albumin serotransferrin and haptoglobin in corona. More experiments are suggested to confirm this finding such as dot blotting assay. Please refer to Figure 4d in Nature Communications | (2024) 15:1159 or Figure 2e in Nature Nanotechnology | Volume 18 | September 2023 | 1067–1077.

8. The last two summary paragraphs are suggested to be integrated into one.

9. The study should also discuss the limitations and potential caveats of this method, such as selectivity of small molecules, requirements for plasma sample preprocessing, possible variability between samples and the effects of the spiking of small molecules on the variables of proteome data. It is important to acknowledge these limitations and suggest strategies to mitigate their impact on the reliability and accuracy of proteome analysis.

Version 1:

Reviewer comments:

Reviewer #1

(Remarks to the Author)

The authors have now adequately addressed the concerns raised on previous submission. The revised manuscript is acceptable to me.

Reviewer #2

(Remarks to the Author)

Thank you for the revised manuscript and your detailed responses to my previous concerns. While I appreciate the additional experiments and clarifications provided, there are still several issues that prevent the manuscript from being acceptable for publication. These issues can be grouped into two main areas:

1. Mechanistic Understanding and Selection of PtdChos:

Although the revised manuscript offers some insights (via the addition of computational analysis) into the interaction between albumin and PtdChos, the fundamental mechanistic understanding remains insufficient. It is still unclear whether PtdChos interacts with the nanoparticle (NP) surface in the presence of plasma proteins. If the goal is to optimize a plasma analysis pipeline using PtdChos alone, the authors should focus on determining whether PtdChos interacts with the NP surface in the presence of plasma proteins. Control experiments that assess colloidal stability by incubating PtdChos with NPs in the absence of proteins are irrelevant. The authors could clarify this by performing size exclusion chromatography to separate corona-coated NPs from PtdChos complexes. Additionally, how confident are the authors that centrifugation effectively removes PtdChos complexes? Additional control experiments focusing on one type of NP and PtdChos should be conducted.

2. Biological Variability and Technical Repeatability:

The use of plasma from only four donors does not sufficiently address biological variability. Although the revised manuscript includes coefficients of variation and additional experiments, the technical repeatability of the proteomics workflow remains a significant concern. Both the number of biological replicates (from individual donors) and technical replicates (using pooled plasma sources) should be expanded to provide a more robust assessment of the pipeline's reproducibility.

Reviewer #3

(Remarks to the Author)

Version 2:

Reviewer comments:

Reviewer #2

(Remarks to the Author)

N/A

Reviewer #3

(Remarks to the Author)

REVIEWER COMMENTS

Reviewer #1:

The work presented in the submission titled, "Deep Plasma Proteome Profiling by Modulating Single Nanoparticle Protein Corona with Small Molecules" has a clearly defined hypothesis and definitely present some very interesting results. However, the authors are requested to work on the following suggestions to further revise the manuscript.

1. Abstract: The authors may include the specifics of small biomolecules here if possible. Also please indicate the specific concentration of phosphatidylcholine.

Authors' response: The specifics of small molecules are now added to the abstract. The optimal PtdChos concentration of 1000 µg/ml is now also added to the abstract.

2. Introduction: In the second paragraph, can the authors cite limitations of specific methods separately. What about resolution in terms of no. of quantifiable proteins in the alternate existing methods.

In the third para, what about any specific size criteria and composition, do they generalize the nano sized particles for single NP strategy here?

Authors' response: We have expanded the discussion on the limitations of the strategies mentioned and provided additional references that address further constraints in the depletion techniques.

Regarding the third paragraph: While the physicochemical properties of nanoparticles do indeed influence the structure of their protein corona, it is generally observed that nanoscale materials exhibit different protein abundances compared to the original plasma protein composition. In essence, most nanoparticles (except those with protein-repellent coatings) have the potential to form a protein corona with a distinct protein composition and abundance, differing from the native plasma proteins. We have mentioned this matter in the paragraph three of the introduction in the revised manuscript.

In the Last paragraph of the introduction, can the authors clarify more on the basis of selection of small molecules. Is it based on previous findings or preliminary results.

Authors' response: We have added the basis for selection of small molecules into the last paragraph of the introduction. In general, the selection was based on the small molecules' capability to interact with proteins which, in turn, could change the plasma protein interactions with nanoparticles and create new protein corona profiles.

Any specific rationale for selecting polystyrene NPs in this work. Can this advantage of small molecule-based analysis be leveraged for other types of particles having similar size range or composition.

Authors' response: We specifically chose highly uniform polystyrene NPs for the following two main reasons: i) their protein corona encompasses a broad spectrum of protein categories such as immunoglobulin, lipoproteins, tissue leakage proteins, acute phase proteins, complement proteins, and coagulation factors. This diversity is critically important for biomarker identification; ii) we have performed rigorous optimization using a wide range of characterizations including mass spectrometry to analyze the pure protein corona of these NPs, ensuring highly accurate

and reproducible results (see our recent *Nature Commun.*^{1,2,3} papers for details). These factors make polystyrene NPs particularly suited for our objectives in this research.

3. Results:

In the first paragraph, how the authors select the composition of molecular sauce 1 and 2?

Authors' response: We used combinations of vitamins, nutrition, lipid, and metabolome in each category to see whether their combinations could potentiate the effect on improving the depth of proteome coverage.

In the second para, whether charge of the bare NPs (Zeta potential) had any role to play in protein corona formation. Whether the pre incubation time of small molecules with the plasma proteins was optimized and determined at 1 h. Similarly, the NP concentration of 0.2 mg/ml, how this was set? Since the authors presented a method development here, the steps need more clarity. Through they mentioned, "These methodological parameters were refined from previous studies", please indicate specific details.

Authors' response: Numerous studies revealed that the charge of the nanoparticles have direct effect in the composition of protein corona (please see our recent meta-analysis of the protein corona literature on the role of physicochemical properties of nanoparticles and protein corona for details: <https://onlinelibrary.wiley.com/doi/10.1002/smll.202301838>). For this research, the main objective is to create uniform protein corona and we chose the polystyrene nanoparticles based on our extensive experience in standardizing the experimental condition to obtain robust protein corona data. The reason for incubating the small molecules with proteins for 1 hour was to give them enough time to interact with plasma proteins. With regards to the nanoparticles concentration, we chose this concentration to avoid formation of protein contamination in protein corona (please see the results of our mechanistic study in defining protein contamination and strategies to overcome them in the following Nature Commun papers: <https://www.nature.com/articles/s41467-020-20884-9> and <https://www.nature.com/articles/s41467-021-27643-4>). We have now added this information in the revised manuscript.

The last line of the section titled, "Small molecules diversify the protein corona composition" can be moved to discussion.

Authors' response: Done.

4. Discussion:

Can the authors clarify on influence of competitive binding of proteins and selected small molecules to the NPs surface, and resolution of overall composition of protein in the corona?

Authors' response: We have conducted a molecular dynamic analysis and defined the interactions of small molecules with albumin and how it can further reduce the albumin affinity to the surface of nanoparticles. See Figure 4 of the revised manuscript for details.

The authors stated, "the simple addition of PtdChos to plasma can significantly reduce albumin's affinity for the surface of polystyrene NPs...". Whether relative affinity (binding constant) has a role to play here.

Authors' response: Our molecular dynamics simulation analysis corroborates existing literature, confirming that PtdChos can bind to albumin. Moreover, our findings demonstrate that the increased proteomics depth observed when using NPs in conjunction with PtdChos is consistent across different types of nanoparticles. This suggests that the enhanced proteome coverage is primarily due to the reduced binding of albumin to the nanoparticle surface, which is facilitated by its interaction with PtdChos.

The results and analytical potential notwithstanding can the authors comment on the following points

(i) role of the competitive binding of small molecules and proteins from a mixture onto the NP surface, (ii) effect of pH on the analytical process, (iii) whether this can be generalized to the other NPs and finally whether the size of the NPs have any specific role to play in the process, (iv) There are several studies reporting conformational changes in the plasma proteins by NPs (PS NPs too). Whether this would impact the analytical potential to develop newer diagnostics approaches?

Authors' response: (i) we have now added a discussion around this raised point in the revised manuscript. (ii) We did not investigate the effect of pH, as most plasma proteomics and NP corona studies are performed at neutral pH to avoid protein precipitation. (iii), we have now investigated the impact of size and charge on the role of PtdChos in improving proteome coverage, the results of which are summarized in **Supplementary Figure 10**. (iv) The reviewer is correct that the binding of proteins to NP surface might change their conformation. In the past, we have tried to apply proteome-wide methods such as Thermal Proteome Profiling and Proteome Integral Solubility Alteration (PISA) assay to monitor these structural changes. However, in neat plasma, the variability in such datasets is too high for obtaining meaningful and statistically significant results. And diluting the plasma prior to such analysis would not be physiologically relevant. If performed on NPs with protein corona, the material is simply not enough for such techniques. Therefore, the community is technologically limited at the moment to address this question across the whole proteome and currently, this can only be performed for individual proteins.

Conclusions: The authors are encouraged to add a few lines on analytical challenges and limitations of the method in the conclusions section.

Authors' response: We have now added the analytical challenges and limitations of our method in the conclusion section.

Reviewer #2

In this article, Ashkarran et al. aim to build on prior research by exploring the utilization of the nanoparticle biomolecule corona for deep plasma proteomics analysis. The authors investigate how the addition or spiking of small molecules in plasma samples influences corona formation and subsequently impacts the number of plasma proteins identified by mass spectrometry. While the hypothesis presented in this work holds some novelty, there are major concerns and substantial experimental and data analysis work required for the manuscript to be considered for publication. Furthermore, given the primary focus on developing a proteomics pipeline without direct implications for biomarker discovery, the manuscript might be more suitable for publication in a proteomics-focused journal. Therefore, I would not recommend this manuscript for publication in Nature Communications.

Authors' response: We appreciate the reviewer's insightful comments. In response, we have conducted additional complementary experiments using various types of nanoparticles and employed top-down proteomics to identify some FDA-approved biomarkers. We also performed

molecular dynamic simulations to further strengthen the robustness of our claims. We would like to highlight that the key contribution of this paper lies in significantly enhancing the depth of proteome coverage within a single plasma sample.

Major Concerns:

- The underlying mechanism driving the hypothesis of this work remains vague, as the authors have not sufficiently addressed or experimentally investigated it. It is unclear whether the observed increase in the number of identified proteins is a result of the interaction between small molecules and the NP surface or a result of the competitive interaction of proteins with small molecules and NP surfaces. It is essential to understand whether small molecules interact with plasma proteins, with the NP surface, or potentially with both. Does for example PtdChos interact with albumin and other proteins in blood and subsequently reduces the interaction of highly abundant proteins with the NP surface? If this interaction occurs, the authors should provide evidence demonstrating the removal of PtdChos:protein complexes upon purification of the corona-coated NPs. Clarifying these aspects is crucial for establishing a robust mechanistic understanding and ensuring the validity of the hypothesis proposed by the authors.

Authors' response: We have now addressed this issue in the revised manuscript, clarifying that the observed increase in the number of quantified proteins is primarily due to the interactions of small molecules with proteins which, in turn, alter proteins interactions with nanoparticles. Our dynamic simulation analysis (please see **Figure 4** of the revised manuscript for details), consistent with existing literature, clearly identifies multiple binding sites on albumin for PtdChos. This binding effectively hinders albumin's interaction with the NP surface. Additionally, we found that directly adding PtdChos to nanoparticles in the absence of plasma proteins caused significant nanoparticle aggregation, compromising their colloidal stability.

Moreover, our new top-down proteomics analysis revealed a distinct PtdChos peak in the mass spectrometry outcomes, confirming its direct interaction with proteins, as hypothesized in the manuscript (please see **Figure 3** of the revised manuscript for details). These new experiments and computational analyses collectively validate our hypothesis that the selected small molecules interact with proteins, altering their binding profiles with nanoparticles. This results in the formation of diverse protein corona profiles, consistent with current literature.

While further exploration of the underlying mechanisms is indeed of interest, it is beyond the primary scope of this study. Investigating the removal of the complex, though intriguing, is challenging, given that much less than 1% of proteins bind to the NP surface, leading to only minimal changes in the unbound fraction of albumin.

- Selection of small molecules and 'personalized corona'. How were the types of small molecules and their combinations ('sauce') selected? Additionally, a significant limitation of this work is the use of pooled healthy plasma as the plasma source (three technical replicates). This experimental design fails to address individual variations in the plasma proteome. For instance, what if glucose levels are inherently elevated in an individual's plasma? Therefore, biological replicates (plasma samples obtained from various individuals) are necessary to demonstrate the efficiency the repeatability of the workflow proposed.

Authors' response: The selection of the small molecules was mainly based on their class (metabolites, lipids, vitamins, and nutrients) and a review of the literature to identify metabolites

that can bind to different plasma proteins. We also used combinations of vitamins, nutrition, lipid, and metabolome in each category to see whether their combinations potentiate the effects on improving the depth of proteome coverage. This statement is now added to the introduction and now reads: "The selection of these molecules was based on their ability to interact with a broad spectrum of proteins, which significantly influences the composition of the protein corona surrounding NPs. For example B complex components can interact with a wide range of proteins including albumin^{4, 5}, hemoglobin⁴, myoglobin⁶, pantothenate permease⁷, acyl carrier protein⁸, lactoferrin⁹, prion¹⁰, β -amyloid precursor¹¹, and niacin-responsive repressor¹². Additionally, to assess the collective effects of these molecules, we analyzed two representative "molecular sauces." Molecular sauce 1 contained a blend of glucose, triglyceride, diglycerol, PtdChos, and molecular sauce 2 consisted of PE, PtdIns, IMP, and vitamin B complex."

We appreciate the reviewer's insight on the second point and have addressed this by incorporating experiments with plasma from different donors, using NPs of varying sizes and surface charges. Our results demonstrate that the effect of PtdChos on increasing the number of quantified plasma proteins is consistent across these different conditions. As shown in the results, when additional individual plasmas were analyzed, the number of identified proteins increased, primarily reflecting the natural variations in proteomics profiles among different individuals. This consistency underscores the robustness of our findings across diverse biological samples and nanoparticle types.

- The technical repeatability of the developed proteomics workflow is a significant concern in this work. Authors conduct only three technical replicates from pooled plasma, which may not be sufficient to assess the variability adequately. Calculating the coefficient of variation is essential to demonstrate the repeatability of the pipeline before proceeding to biological replicates.

Authors' response: We have now reported all CVs of the experiments in Supplementary Tables 1-3. We also provided the results of additional experiments on various types of nanoparticles and additional individual plasmas showing the excellent reproducibility of the outcomes.

- This work could have been significantly strengthened by incorporating additional experimental analyses using patient samples for plasma analysis. Such inclusion would have illustrated the advantages of the suggested workflow over previously published methodologies.

Authors' response: Additional studies and experiments using a wide range of diseased plasmas is the focus of our ongoing projects for biomarker discovery purposes. Since the identified biomarkers should be validated, we believe these will become part of a separate and more focused publication. However, in the revised manuscript we designed new experiments and used plasmas from four different individual donors. In addition, we used to-down proteomics to prove that the addition of small molecules to the plasma can increase the numbers of quantified disease associated proteins.

- One of the main findings in this work is the utilization of PtdChos to mitigate the interaction of NPs with highly abundant proteins, particularly albumin. It is not clear if authors suggest using PtdChos alone or in combination with other small molecules. In Figure 1a PtdChos-treated samples have the highest number of identified proteins. Could authors explain why it works better alone rather than in combination with other small molecules ('Molecular sauce 1')? Why is it important to discuss the cumulative number of unique identified proteins? Do authors suggest to

aliquot plasma samples and separately analyse them with different small molecules? That would increase the steps/time/plasma volume required.

Authors' response: When designing the experiment, our major aim was to enhance the plasma proteome coverage by using various small molecules. A similar approach is used by employing an array of NPs. However, in light of the results obtained with PtdChos alone, we are now reinforcing the individual application of this small molecule in plasma proteomics. This issue has now been clarified in the discussion.

- Page 4, Authors mention: 'We note that our analysis focuses on the fold changes of quantified proteins compared to plasma and the untreated corona, rather than the absolute numbers of quantified proteins. This is mainly because the numbers of quantified proteins are strongly dependent to the workflow of mass spectrometry'.

I find this statement confusing and potentially misleading. While I agree that the number of identified proteins depends on the instrument settings and data analysis workflow used, the authors throughout the manuscript primarily focus the discussion on the number of identified proteins in comparison to previous work. Moreover, they employ two different experimental settings, instruments/facilities, digestion protocols, and data analysis workflows (Proteome Discoverer and R), which, in my opinion, weaken rather than strengthen the manuscript. Additionally, considering the emphasis given on the authors' previous work regarding mass spectrometry workflow biases, it would be beneficial for the authors to provide more clarity about the data analysis performed. The current paragraph on data analysis provides minimal information, and I believe more detail is necessary. Furthermore, it is unclear why only one replicate was performed for the DIA analysis.

Authors' response: The validation of this statement was previously demonstrated in our earlier study, where different core facilities analyzing identical protein corona samples reported a wide range of protein identifications, varying from approximately 200 to over 1,300 proteins.² The significant impact of mass spectrometry workflow on proteomics outcomes is well-documented and thoroughly understood in both bottom-up and top-down proteomics fields.^{13, 14, 15, 16, 17} Our research further underscores the severity of this issue in the context of protein corona studies.^{2, 3, 18} To enhance clarity and avoid any potential confusion, we have revised our statement as follows: "Mass spectrometry workflow and the type of data analysis have a critical influence on proteomics outcomes in general^{13, 14, 15, 16, 17}, as well as in the specific field of protein corona research^{2, 3, 18}. For instance, our recent study demonstrated that identical corona-coated polystyrene NPs analyzed by different mass spectrometry centers resulted in a wide range of quantified proteins, varying from 235 to 1,430.² To mitigate the impact of these variables on the interpretation of how small molecules can enhance proteome coverage, we chose to report our data as fold changes in the number of quantified proteins relative to control plasma and untreated corona samples, all while using the same mass spectrometry workflow and data analysis methods. This approach offers a more robust and objective assessment of the role of small molecules in enhancing proteome analysis, minimizing the confounding effects of different workflows and data analysis techniques that may be employed by various researchers."

The deliberate selection of different facilities and workflows in this manuscript was intended to showcase that PtdChos can consistently enhance plasma proteome coverage across various LC-MS workflows and search engines. This demonstrates the broad applicability and robustness of our approach.

As requested by the reviewer, we have now added more details to the data analysis section.

Minor Concerns/comments:

- While employing a single type of NP can offer advantages over utilizing multiple types of NP surface modifications, the authors should discuss why this specific type of NP was chosen. For example, is there a particular reason for using negatively charged NPs?

Authors' response: The use of single NPs offers several advantages over multiple NPs, particularly in terms of commercialization and the regulatory complexities associated with multi-NP systems. Additionally, utilizing a single type of nanoparticle can streamline the mass spectrometry analysis process, reducing the time required to analyze large cohorts in plasma proteomics studies.

Why do we use polystyrene NPs?

Our team has extensive experience in analyzing the composition and profiles of the protein corona on various types of nanoparticles, including gold^{19, 20, 21}, superparamagnetic iron oxide^{22, 23, 24}, graphene oxide^{25, 26, 27}, iron-platinum²⁸, zeolite^{29, 30}, silica^{31, 32}, polystyrene^{1, 2, 3, 31}, silver³³, and lipids^{34, 35, 36}. In this study, we specifically selected highly uniform polystyrene nanoparticles for two primary reasons:

Comprehensive Protein Coverage: Polystyrene NPs have a protein corona that encompasses a broad spectrum of protein categories, including immunoglobulins, lipoproteins, tissue leakage proteins, acute phase proteins, complement proteins, and coagulation factors. This diversity is crucial for achieving wide proteome identification, which is essential for our research objectives.

Optimized and Reproducible Analysis: these particles are tested widely for numerous applications in nanobiomedicine: we^{1, 2, 3} and other groups^{37, 38, 39, 40, 41} have conducted extensive optimization, employing a wide range of characterizations, including mass spectrometry, to analyze the pure protein corona of polystyrene NPs. This rigorous optimization ensures highly accurate and reproducible results.

In response to reviewer feedback, we have also included experiments using silica and polystyrene NPs with various sizes and surface charges to demonstrate that the addition of small molecules enhances proteome coverage across different types of nanoparticles. This approach further validates the robustness and versatility of our method across diverse nanoparticle platforms.

- Why were plasma samples diluted before protein corona formation? Authors should include dilution experiments, including the analysis of protein corona samples formed in undiluted plasma, to provide a comprehensive understanding of the effects of dilution on corona formation.

Authors' response: We used this concentration to align with established protocols⁴² for protein corona analysis. Specifically, we selected a 55% concentration of human plasma to mirror the physiological conditions in blood, where approximately 55% of the volume consists of plasma, while the remaining 45% is composed of red blood cells, white blood cells, and platelets suspended within the plasma. This approach ensures that our studies closely replicate the natural environment, providing more relevant and accurate insights into protein corona formation.

- While authors discuss some previous studies that utilize the NP biomolecule corona formation for proteomics analysis of plasma samples, they have neglected to cite relevant previously published work on this topic.

Authors' response: We have now cited other publications in the field that we are aware of. We appreciate it if the reviewer can provide details of additional relevant references that we might have missed.

- In the introduction/discussion of the manuscript, the authors mention that plasma depletion methodologies are costly and time-consuming. How does the suggested workflow compare to immunodepletion methodologies in terms of cost and time efficiency? Corona formation and purification is also a multistep protocol.

Authors' response: This is mainly because of the available automation strategies in the field of protein corona and the capacity of this single nanoparticle strategy to improve proteomics analysis of multiple cohort samples in a short period of time. Details on the benefits of protein corona proteomics over other depletion strategies are summarized in detail elsewhere^{43, 44}.

- Page 4, Authors mention: 'Interestingly, the concentration of small molecules did not significantly affect the number of quantified proteins; only a small stepwise reduction in the number of quantified proteins was noted with increasing concentrations of glucose and diglycerol'. Could you please clarify where this data is presented?

Authors' response: The data is presented in **Figure 1a**.

- In my opinion, KEGG analysis does not add any important conclusion to this manuscript as healthy pulled plasma was used; I would understand it if plasma samples obtained from different individuals were analysed.

Authors' response: We partially agree with the reviewer and understand the concern. Our intention was to demonstrate the feasibility of performing pathway analysis in the context of similar proteome studies on patient samples. To support this, we have included these analyses in the supplementary materials as an example of how such an approach could be applied in future studies.

Reviewer #3

In this manuscript, Ashkarran et al. reported an intriguing strategy to enhance the plasma proteome coverage through LC-MS/MS analysis. By incorporating small molecules into plasma, they significantly increased the diversity of protein coronas formed on nanoparticles (NPs), subsequently elevating the sensitivity for detecting proteins with lower abundances. This innovative and efficient methodology offers profound implications for deepening plasma proteome analysis, particularly in the realm of clinical biomarker discovery for disease diagnosis and monitoring. Nevertheless, several concerns ought to be addressed prior to publication:

Authors' response: We thank the reviewer for their positive assessment of our work.

1. The novelty of the study lies in using a single NP type and a small molecule to analyze a single plasma sample, setting a new standard in proteomic depth. However, to further enhance the rigor and reliability of the study, it is suggested that the authors provide more detailed information on the biological significance of the small molecules selected and the specific mechanisms of their interactions with proteins.

Authors' response: We thank the reviewer for this constructive comment. The following paragraph has been added to the revised version with regards to the importance of the small molecule selection and their mechanism of actions in altering protein corona profile: "The selection of these molecules was based on their ability to interact with a broad spectrum of proteins, which significantly influences the composition of the protein corona surrounding NPs. For example B complex components can interact with a wide range of proteins including albumin⁴.

⁵, hemoglobin⁴, myoglobin⁶, pantothenate permease⁷, acyl carrier protein⁸, lactoferrin⁹, prion¹⁰, β -amyloid precursor¹¹, and niacin-responsive repressor¹². Additionally, to assess the collective effects of these molecules, we analyzed two representative "molecular sauces." Molecular sauce 1 contained a blend of glucose, triglyceride, diglycerol, PtdChos, and molecular sauce 2 consisted of PE, PtdIns, IMP, and vitamin B complex."

2. In this study, the majority of the smaller molecules employed were biological metabolites, chosen primarily due to their capacity to interact with plasma proteins. Consequently, it is expected that these metabolites would naturally be present in plasma. However, a crucial consideration lies in the background concentrations of these metabolites within the plasma. Did the authors factor in these endogenous concentrations when determining the pre-designed spiking concentrations? Upon examination of the results, it appears that the concentrations of these metabolites did not significantly influence the number of proteins identified for most molecules. This observation deserves further exploration and discussion, as it may provide insights into the complexity of interactions between metabolites and plasma proteins.

Authors' response: We appreciate the reviewer's insightful feedback and have expanded the discussion to address the raised point. Given that these experiments are conducted *ex vivo* for proteomics purposes, the relative concentration of the used small molecules in plasma proteins is not a critical factor in our study. Instead, our focus was on selecting a broad range of small molecule concentrations to determine the optimal levels for maximizing proteome coverage. While some small molecules did not show significant changes in proteome coverage across the tested concentrations, we observed that for PtdChos, the concentration had a notable impact, with the most pronounced effects occurring at 1000 $\mu\text{g}/\text{ml}$.

3. The concentrations of each molecules in the two "molecular sauces" should be specified.

Authors' response: The concentration of each small molecule was carefully adjusted to ensure that the final concentration in the combined molecular solutions were 10, 100, or 1000 $\mu\text{g}/\text{ml}$ for each component, consistent with the concentration used for individual small molecules. We have now included additional details on this adjustment process in the revised manuscript.

4. The study found that the addition of 1000 $\mu\text{g}/\text{ml}$ phosphatidylcholine (PtdChos) significantly increased the number of unique proteins within the protein corona (897 proteins). This finding has significant clinical translation potential. However, it is suggested that the authors further explore the impact of PtdChos concentration on other plasma proteins and whether this selective depletion mechanism affects the detection sensitivity of low-abundance proteins. More additional bioinformatics analysis on PtdChos-treated plasma samples were recommended to determine whether these changes affect the identification of potential biomarkers.

Authors' response: The impact of PtdChos on quantification of low abundant proteins is clear in Fig 1a and 1c as well as in Figure 2. As noted in our hierarchical clustering analysis in Figure 1c, there is a cluster of proteins that are only present in the presence of PtdChos, which would otherwise not be quantified.

5. In the DDA analysis, the number of identified proteins in untreated corona and ptdchos (100 $\mu\text{g}/\text{uL}$) group were 681 and 474 (Figure 1a). In the DIA analysis, their counterparts were 617 and 836 (Figure 2c). Why DIA analysis could greatly improve the depth of identified proteins for ptdchos group but not for untreated corona group?

Authors' response: These experiments were performed independently, and variabilities are therefore expected. Further variabilities are also introduced due to the LC-MS workflows used, as we have discussed in the past^{2, 18}. This is why we focus on the fold change of the number of quantified proteins in NP+PtdChos group to the NP group alone (and not just the protein count).

6. More details should be provided on the optimization of the data-independent acquisition (DIA) method, including the selection of acquisition parameters, data processing workflows, and how to ensure the accuracy and reproducibility of the data. Furthermore, authors should discuss the potential application of this method in different types of plasma samples and whether the detection of specific proteins can be optimized by adjusting the concentration of small molecules.

Authors' response: We used the method that was routinely used in Biozentrum core facility for plasma proteome analysis. The experimental method details are now added to the materials and methods. According to the reviewer's comment, we have now added a discussion on how our strategy can be used for detection of specific proteins tailored for different diseases and this now reads: "This ability of small molecules to modify the protein composition on NPs highlights their potential for early disease diagnosis (e.g., apolipoproteins in cardiovascular and neurodegenerative disorders)^{45, 46}, where these protein categories are crucial in disease onset and progression⁴⁵."

7. Proteomics analysis showed that Ptdchos efficiently reduce the abundance of albumin serotransferrin and haptoglobin in corona. More experiments are suggested to confirm this finding such as dot blotting assay. Please refer to Figure 4d in Nature Communications | (2024) 15:1159 or Figure 2e in Nature Nanotechnology | Volume 18 | September 2023 | 1067–1077.

Authors' response: We believe that LC-MS data is more robust than such complementary techniques, as it provided peptide level and isoform level information. However, we have now included a statement citing this paper.⁴⁷

8. The last two summary paragraphs are suggested to be integrated into one.

Authors' response: Done.

9. The study should also discuss the limitations and potential caveats of this method, such as selectivity of small molecules, requirements for plasma sample preprocessing, possible variability between samples and the effects of the spiking of small molecules on the variables of proteome data. It is important to acknowledge these limitations and suggest strategies to mitigate their impact on the reliability and accuracy of proteome analysis.

Authors' response: The limitations are now discussed in the conclusions and now reads "We anticipate that this platform will see extensive use in plasma proteome profiling, offering significant potential for advancements in disease diagnostics and monitoring. However, one critical challenge that must be addressed is the standardization of proteomics analysis of the protein corona. Ensuring consistent and reproducible results across laboratories and core facilities is essential for the rapid development and successful translation of this platform into clinical applications.^{2, 3, 18} Addressing this challenge will require coordinated efforts from the scientific community to establish robust, universally accepted protocols. There are a few additional foreseeable limitations with the application of PtdChos. In certain scenarios, any depletion strategy could lead to distortion of the abundance of proteins in plasma, which can be mitigated by enforcing proper controls. Moreover, upon discovery of a biomarker, it can be validated in the cohort using orthogonal techniques such as Western blotting. Furthermore, similar to other albumin depletion strategies, certain proteins bound to albumin might be co-depleted (albuminome)⁴⁸."

References:

1. Sheibani S, Basu K, Farnudi A, Ashkarran A, Ichikawa M, Presley JF, *et al.* Nanoscale characterization of the biomolecular corona by cryo-electron microscopy, cryo-electron tomography, and image simulation. *Nature communications* 2021, **12**: 573.
2. Ashkarran AA, Gharibi H, Voke E, Landry MP, Saei AA, Mahmoudi M. Measurements of heterogeneity in proteomics analysis of the nanoparticle protein corona across core facilities. *Nature Communications* 2022, **13**(1): 6610.
3. Gharibi H, Ashkarran AA, Jafari M, Voke E, Landry MP, Saei AA, *et al.* A uniform data processing pipeline enables harmonized nanoparticle protein corona analysis across proteomics core facilities. *Nature Communications* 2024, **15**(1): 342.
4. Fonda ML. Vitamin B6 metabolism and binding to proteins in the blood of alcoholic and nonalcoholic men. *Alcoholism: Clinical and Experimental Research* 1993, **17**(6): 1171-1178.
5. Panja S, Khatua DK, Halder M. Simultaneous Binding of Folic Acid and Methotrexate to Human Serum Albumin: Insights into the Structural Changes of Protein and the Location and Competitive Displacement of Drugs. *ACS Omega* 2018, **3**(1): 246-253.
6. Ghosh R, Thomas DS, Arcot J. Molecular Recognition Patterns between Vitamin B12 and Proteins Explored through STD-NMR and In Silico Studies. *Foods* 2023, **12**(3): 575.
7. Jackowski S, Alix J-H. Cloning, sequence, and expression of the pantothenate permease (panF) gene of Escherichia coli. *Journal of bacteriology* 1990, **172**(7): 3842-3848.
8. Musa TL, Ioerger TR, Sacchettini JC. The Tuberculosis Structural Genomics Consortium: A Structural Genomics Approach to Drug Discovery. In: Joachimiak A (ed). *Advances in Protein Chemistry and Structural Biology*, vol. 77. Academic Press, 2009, pp 41-76.
9. Adhel E, Ha Duong N-T, Vu TH, Taverna D, Ammar S, Serradji N. Interaction between carbon dots from folic acid and their cellular receptor: a qualitative physicochemical approach. *Physical Chemistry Chemical Physics* 2023, **25**(20): 14324-14333.

10. Lonsdale D. Thiamin and protein folding. *Medical Hypotheses* 2019, **129**: 109252.
11. Mkrtchyan G, Aleshin V, Parkhomenko Y, Kaehne T, Luigi Di Salvo M, Parroni A, *et al.* Molecular mechanisms of the non-coenzyme action of thiamin in brain: biochemical, structural and pathway analysis. *Scientific Reports* 2015, **5**(1): 12583.
12. Lee DW, Park YW, Lee MY, Jeong KH, Lee JY. Structural analysis and insight into effector binding of the niacin-responsive repressor NiaR from *Bacillus halodurans*. *Scientific Reports* 2020, **10**(1): 21039.
13. Cassidy L, Kaulich PT, Maaß S, Bartel J, Becher D, Tholey A. Bottom-up and top-down proteomic approaches for the identification, characterization, and quantification of the low molecular weight proteome with focus on short open reading frame-encoded peptides. *PROTEOMICS* 2021, **21**(23-24): 2100008.
14. Ignjatovic V, Geyer PE, Palaniappan KK, Chaaban JE, Omenn GS, Baker MS, *et al.* Mass spectrometry-based plasma proteomics: considerations from sample collection to achieving translational data. *Journal of proteome research* 2019, **18**(12): 4085-4097.
15. Picotti P, Aebersold R. Selected reaction monitoring-based proteomics: workflows, potential, pitfalls and future directions. *Nat Methods* 2012, **9**(6): 555-566.
16. Roberts DS, Loo JA, Tsybin YO, Liu X, Wu S, Chamot-Rooke J, *et al.* Top-down proteomics. *Nat Rev Methods Primers* 2024, **4**(1).
17. Yates JR, Ruse CI, Nakorchevsky A. Proteomics by Mass Spectrometry: Approaches, Advances, and Applications. *Annual Review of Biomedical Engineering* 2009, **11**(Volume 11, 2009): 49-79.
18. Ashkarran AA, Gharibi H, Modaresi SM, Saei AA, Mahmoudi M. Standardizing Protein Corona Characterization in Nanomedicine: A Multicenter Study to Enhance Reproducibility and Data Homogeneity. *Nano Letters* 2024, **24**(32): 9874-9881.
19. Mahmoudi M, Lohse SE, Murphy CJ, Fathizadeh A, Montazeri A, Suslick KS. Variation of protein corona composition of gold nanoparticles following plasmonic heating. *Nano letters* 2013, **14**(1): 6-12.

20. Saha K, Rahimi M, Yazdani M, Kim ST, Moyano DF, Hou S, *et al.* Regulation of Macrophage Recognition through the Interplay of Nanoparticle Surface Functionality and Protein Corona. *ACS nano* 2016, **10**(4): 4421-4430.
21. Ashkarran AA, Tadjiki S, Lin Z, Hilsen K, Ghazali N, Krikor S, *et al.* Protein Corona Composition of Gold Nanocatalysts. *ACS Pharmacology & Translational Science* 2024, **7**(4): 1169-1177.
22. Askim JR, Mahmoudi M, Suslick KS. Optical sensor arrays for chemical sensing: the optoelectronic nose. *Chemical Society Reviews* 2013, **42**(22): 8649-8682.
23. Mahmoudi M, Shokrgozar MA, Sardari S, Moghadam MK, Vali H, Laurent S, *et al.* Irreversible changes in protein conformation due to interaction with superparamagnetic iron oxide nanoparticles. *Nanoscale* 2011, **3**: 1127-1138.
24. Sakulkhu U, Mahmoudi M, Maurizi L, Salaklang J, Hofmann H. Protein corona composition of superparamagnetic iron oxide nanoparticles with various physico-chemical properties and coatings. *Scientific reports* 2014, **4**: 5020.
25. Mahmoudi M, Akhavan O, Ghavami M, Rezaee F, Ghiasi SMA. Graphene oxide strongly inhibits amyloid beta fibrillation. *Nanoscale* 2012, **4**(23): 7322-7325.
26. Mao H, Chen W, Laurent S, Thirifays C, Burtea C, Rezaee F, *et al.* Hard corona composition and cellular toxicities of the graphene sheets. *Colloids and Surfaces B: Biointerfaces* 2013, **109**: 212-218.
27. Hajipour MJ, Raheb J, Akhavan O, Arjmand S, Mashinchian O, Rahman M, *et al.* Personalized disease-specific protein corona influences the therapeutic impact of graphene oxide. *Nanoscale* 2015, **7**(19): 8978-8994.
28. Rahman M, Laurent S, Tawil N, Yahia LH, Mahmoudi M. Protein-nanoparticle interactions: the bio-nano interface. Springer Science & Business Media; 2013.
29. Rahimi M, Ng E, Bakhtiari K, Vinciguerra M, Awala H, Mintova S, *et al.* Zeolite Nanoparticles for Selective Sorption of Plasma Proteins. *Scientific reports* 2014, **5**: 17259-17259.
30. Laurent S, Ng E-P, Thirifays C, Lakiss L, Goupil G-M, Mintova S, *et al.* Corona protein composition and cytotoxicity evaluation of ultra-small zeolites synthesized from template free precursor suspensions. *Toxicology Research* 2013, **2**(4): 270-279.

31. Hajipour MJ, Laurent S, Aghaie A, Rezaee F, Mahmoudi M. Personalized protein coronas: a “key” factor at the nanobiointerface. *Biomaterials science* 2014, **2**(9): 1210-1221.
32. Ashkarran AA, Gharibi H, Grunberger JW, Saei AA, Khurana N, Mohammadpour R, *et al.* Sex-Specific Silica Nanoparticle Protein Corona Compositions Exposed to Male and Female BALB/c Mice Plasmas. *ACS Bio & Med Chem Au* 2023, **3**(1): 62-73.
33. Ashkarran AA, Ghavami M, Aghaverdi H, Stroeve P, Mahmoudi M. Bacterial effects and protein corona evaluations: crucial ignored factors in the prediction of bio-efficacy of various forms of silver nanoparticles. *Chemical research in toxicology* 2012, **25**(6): 1231-1242.
34. Bigdeli A, Palchetti S, Pozzi D, Hormozi-Nezhad MR, Baldelli Bombelli F, Caracciolo G, *et al.* Exploring Cellular Interactions of Liposomes Using Protein Corona Fingerprints and Physicochemical Properties. *ACS nano* 2016, **10**(3): 3723-3737.
35. Palchetti S, Digiacoimo L, Pozzi D, Peruzzi G, Micarelli E, Mahmoudi M, *et al.* Nanoparticles-cell association predicted by protein corona fingerprints. *Nanoscale* 2016, **8**(25): 12755-12763.
36. Caracciolo G, Safavi-Sohi R, Malekzadeh R, Poustchi H, Vasighi M, Zenezini Chiozzi R, *et al.* Disease-specific protein corona sensor arrays may have disease detection capacity. *Nanoscale Horizons* 2019, **4**(5): 1063-1076.
37. Lunov O, Syrovets T, Loos C, Beil J, Delacher M, Tron K, *et al.* Differential Uptake of Functionalized Polystyrene Nanoparticles by Human Macrophages and a Monocytic Cell Line. *ACS Nano* 2011, **5**(3): 1657-1669.
38. Tenzer S, Docter D, Rosfa S, Wlodarski A, Kuharev J, Rekić A, *et al.* Nanoparticle Size Is a Critical Physicochemical Determinant of the Human Blood Plasma Corona: A Comprehensive Quantitative Proteomic Analysis. *ACS Nano* 2011, **5**(9): 7155-7167.
39. Tonigold M, Simon J, Estupiñán D, Kokkinopoulou M, Reinholz J, Kintzel U, *et al.* Pre-adsorption of antibodies enables targeting of nanocarriers despite a biomolecular corona. *Nature Nanotechnology* 2018, **13**(9): 862-869.

40. Monopoli MP, Åberg C, Salvati A, Dawson KA. Biomolecular coronas provide the biological identity of nanosized materials. *Nature Nanotechnology* 2012, **7**(12): 779-786.
41. Tenzer S, Docter D, Kuharev J, Musyanovych A, Fetz V, Hecht R, *et al.* Rapid formation of plasma protein corona critically affects nanoparticle pathophysiology. *Nature Nanotechnology* 2013, **8**(10): 772-781.
42. Zhang P, Cao M, Chetwynd AJ, Faserl K, Abdolahpur Monikh F, Zhang W, *et al.* Analysis of nanomaterial biocoronas in biological and environmental surroundings. *Nature Protocols* 2024.
43. Blume JE, Manning WC, Troiano G, Hornburg D, Figa M, Hesterberg L, *et al.* Rapid, deep and precise profiling of the plasma proteome with multi-nanoparticle protein corona. *Nature Communications* 2020, **11**(1): 3662.
44. Mahmoudi M, Landry MP, Moore A, Coreas R. The protein corona from nanomedicine to environmental science. *Nature Reviews Materials* 2023, **8**(7): 422-438.
45. Mahley RW. Apolipoprotein E: from cardiovascular disease to neurodegenerative disorders. *Journal of molecular medicine* 2016, **94**: 739-746.
46. Mahmoudi N, Mahmoudi M. Effects of cholesterol on biomolecular corona. *Nature Nanotechnology* 2023, **18**(9): 974-976.
47. Tang H, Zhang Y, Yang T, Wang C, Zhu Y, Qiu L, *et al.* Cholesterol modulates the physiological response to nanoparticles by changing the composition of protein corona. *Nature Nanotechnology* 2023, **18**(9): 1067-1077.
48. Shi T, Zhou JY, Gritsenko MA, Hossain M, Camp DG, 2nd, Smith RD, *et al.* IgY14 and SuperMix immunoaffinity separations coupled with liquid chromatography-mass spectrometry for human plasma proteomics biomarker discovery. *Methods* 2012, **56**(2): 246-253.

REVIEWER COMMENTS

Reviewer #1:

The authors have now adequately addressed the concerns raised on previous submission. The revised manuscript is acceptable to me.

Reviewer #1:

Authors' response: We thank the reviewer for their time and consideration.

Reviewer #2

Thank you for the revised manuscript and your detailed responses to my previous concerns. While I appreciate the additional experiments and clarifications provided, there are still several issues that prevent the manuscript from being acceptable for publication. These issues can be grouped into two main areas:

1. Mechanistic Understanding and Selection of PtdChos:

Although the revised manuscript offers some insights (via the addition of computational analysis) into the interaction between albumin and PtdChos, the fundamental mechanistic understanding remains insufficient. It is still unclear whether PtdChos interacts with the nanoparticle (NP) surface in the presence of plasma proteins. If the goal is to optimize a plasma analysis pipeline using PtdChos alone, the authors should focus on determining whether PtdChos interacts with the NP surface in the presence of plasma proteins. Control experiments that assess colloidal stability by incubating PtdChos with NPs in the absence of proteins are irrelevant. The authors could clarify this by performing size exclusion chromatography to separate corona-coated NPs from PtdChos complexes. Additionally, how confident are the authors that centrifugation effectively removes PtdChos complexes? Additional control experiments focusing on one type of NP and PtdChos should be conducted.

Authors' response: We sincerely thank you for your valuable time and thoughtful consideration of our revised manuscript. Your insightful comments have significantly contributed to enhancing the clarity and robustness of our study.

We believe we have thoroughly addressed the issue concerning the interactions between PtdChos and nanoparticle surfaces in the revised version of the manuscript. In addition to the colloidal stability experiments and computational modeling previously discussed, we also had incorporated top-down proteomics analyses. This advanced technique utilizes intact proteins to analyze the composition of the protein corona through the quantification of proteoforms, providing a more detailed and accurate depiction of the protein-nanoparticle interactions. We were the first group using this technique in protein corona for robust analysis of intact proteins (see <https://pubs.acs.org/doi/full/10.1021/acsnano.4c04675> for details). Our results revealed that plasma treated with PtdChos exhibited a significant signal corresponding to the small molecule after 60 minutes of separation time (Figure 3b). Furthermore, the process of recovering intact proteins from the surfaces of nanoparticles primarily collects proteins from the outer layer of the protein corona (as shown by various techniques in our recent publication <https://pubs.acs.org/doi/full/10.1021/acsnano.4c04675>), as the inner layer is tightly bound to the NP surfaces through various physical and chemical forces (<https://www.nature.com/articles/nmat2442>). Therefore, if PtdChos interacted with the surface of NPs, we would not see its strong signal in the characterized proteins by the top-down

proteomics. This finding validates our hypothesis that small molecules like PtdChos interact with plasma proteins, leading to the observed variations in the protein corona on the surface of nanoparticles.

Regarding your comments on the efficiency of centrifugation for the separation of nanoparticles without PtdChos complex contamination, we adhere to a rigorous standard protocol to ensure the purity and integrity of our samples. Specifically, we confirm that corona-coated nanoparticles are fully dispersed after the centrifugation process and that there is no residual protein contamination or aggregation that could introduce impurities into the protein corona. This protocol involves careful monitoring of particle dispersion, multiple washing steps to remove unbound proteins and small molecules, and validation of the nanoparticle dispersion state post-centrifugation.

The full validation process is detailed in our earlier publication in Nature Communications (<https://www.nature.com/articles/s41467-020-20884-9>), where we describe the methodologies employed to verify nanoparticle dispersion and protein corona purity comprehensively. To further substantiate our current findings, we have included a representative cryo-TEM validation image below from our experiments. This image illustrates the well-dispersed state of the nanoparticles and the absence of aggregation or contamination, confirming the efficacy of our separation and purification processes.

In instances where we observe nanoparticle aggregation or potential sources of protein contamination, we employ alternative separation techniques such as size-exclusion chromatography (<https://www.nature.com/articles/s41467-017-00600-w>) or field-flow fractionation (<https://pubs.acs.org/doi/10.1021/acsbiomedchemau.4c00001>) to ensure the highest sample purity. However, for the current study, our standard centrifugation protocol proved sufficient, and no such issues were detected.

We hope that these additions and clarifications satisfactorily address your concerns. Your feedback has been invaluable in refining our work, and we are grateful for the opportunity to improve our manuscript.

Representative Cryo-TEM images of protein corona-coated polystyrene nanoparticles after centrifugation show the formation of uniform, aggregate-free nanoparticles.

2. Biological Variability and Technical Repeatability:

The use of plasma from only four donors does not sufficiently address biological variability. Although the revised manuscript includes coefficients of variation and additional experiments, the technical repeatability of the proteomics workflow remains a significant concern. Both the number of biological replicates (from individual donors) and technical replicates (using pooled

plasma sources) should be expanded to provide a more robust assessment of the pipeline's reproducibility.

Authors' response: We acknowledge the reviewer's concern regarding the limited number of plasma samples used in our study. However, addressing individual variability falls outside the scope of this work. The primary objective of our study is to demonstrate that small molecules capable of preferentially interacting with highly abundant proteins can reduce their binding to nanoparticle surfaces. This reduction allows low-abundance proteins to participate more significantly in the protein corona profile. Given that the concentrations of the 22 most abundant plasma proteins show minimal variation between individuals, our sample size—which includes one pooled plasma sample and four individual plasma samples tested with various types of nanoparticles—is sufficient to robustly validate our main hypothesis.

The critical challenges associated with individual variability are more pertinent to biomarker discovery studies, where researchers aim to detect the remaining low-abundance proteins (over 15,000 proteins beyond the highly abundant ones). Therefore, future studies intending to apply our strategy for biomarker discovery should indeed utilize a larger cohort of plasma samples to account for individual differences.

Additionally, we have employed the concept of actual causality, as outlined by Halpern and Pearl (<https://direct.mit.edu/books/oa-monograph/3451/Actual-Causality>), to mathematically substantiate how various small molecules—including metabolites, lipids, vitamins, and nutrients—spiked into plasma can induce diverse protein corona patterns based on our proteomics results. Our findings revealed that among the small molecules tested, PtdChos were the actual cause of the observed increase in the proteomic depth of the plasma sample (<https://www.biorxiv.org/content/10.1101/2024.09.10.612356v1>). This effect was achieved by reducing the binding of highly abundant proteins and enhancing the representation of low-abundance proteins on the nanoparticle surfaces.

We have incorporated the above discussion into the revised manuscript to clarify these points.